# LANGUAGE AGENT TREE SEARCH UNIFIES REASONING ACTING AND PLANNING IN LANGUAGE MODELS

## ABSTRACT

While large language models (LLMs) have demonstrated impressive performance on a range of decision-making tasks, they rely on simple acting processes and fall short of broad deployment as autonomous agents. We introduce LATS (Language Agent Tree Search), a general framework that synergizes the capabilities of LLMs in planning, acting, and reasoning. Drawing inspiration from Monte Carlo tree search commonly used in model-based reinforcement learning, LATS employs LLMs as agents, value functions, and optimizers, repurposing their latent strengths for enhanced decision-making. What is crucial in this method is the use of an environment for external feedback, which offers a more deliberate and adaptive problem-solving mechanism that moves beyond the limitations of existing techniques. Our experimental evaluation across diverse domains, such as programming, HotPotQA, and WebShop, illustrates the applicability of LATS for decision-making while maintaining competitive reasoning performance. In particular, LATS achieves 94.4% for programming on HumanEval with GPT-4 and an average score of 75.9 for web browsing on WebShop with GPT-3.5, demonstrating the effectiveness and generality of our method.

## 1 INTRODUCTION

General autonomous agents capable of reasoning and decision-making in a variety of environments (Wooldridge & Jennings, 1995) have been of longstanding interest in the field of artificial intelligence. While this has traditionally been studied in reinforcement learning, the recent rise of large language models (LLMs) (Brown et al., 2020; Chowdhery et al., 2022; Touvron et al., 2023; OpenAI, 2023) with strong reasoning and general adaptability offers an alternative paradigm. Not only have LLMs excelled on standard NLP tasks such as text summarization (Nallapati et al., 2016) or natural language inference (Bowman et al., 2015), but they have been adapted to an increasingly diverse set of tasks that often require advanced common-sense reasoning or quantitative skills (Cobbe et al., 2021; Saparov & He, 2022). LLMs are also capable of performing in complex environments that involve knowledge and reasoning, such as web navigation (Yao et al., 2022; Deng et al., 2023), tool-use (Schick et al., 2023), or open-ended games (Fan et al., 2022).

Reasoning and acting abilities have also been improved by prompting techniques that augment LLMs with feedback or observations from an external environment (Yao et al., 2023b; Gao et al., 2022; Shinn et al., 2023). This eliminates the need to rely entirely on the base abilities of the Language Model (LM), enhancing it through external tools or semantic feedback. Despite this strength, these methods are reflexive and fall short of humans' deliberate and thoughtful decision-making characteristics to solve problems (Sloman, 1996; Evans, 2010). In particular, such methods fail to consider multiple reasoning paths or to plan ahead. Recent search-guided LLM works (Xie et al., 2023; Yao et al., 2023a; Hao et al., 2023) address this

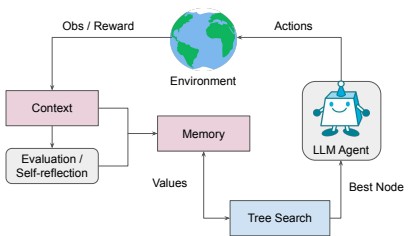

Figure 1: An overview of LATS. LATS uses an external environment and self-reflection to improve reasoning and decision-making. Prior states, values, and reflections are stored in external memory.

| Approach | Reasoning | Acting | Planning | Self Reflection | External Memory |
|---|---|---|---|---|---|
| CoT (Wei et al., 2022) | ✓ | ✗ | ✗ | ✗ | ✗ |
| ReAct (Yao et al., 2023b) | ✓ | ✓ | ✗ | ✗ | ✗ |
| ToT (Yao et al., 2023a) | ✓ | ✗ | ✓ | ✓ | ✓ |
| RAP (Hao et al., 2023) | ✓ | ✗ | ✓ | ✗ | ✓ |
| Self-Refine (Madaan et al., 2023) | ✓ | ✗ | ✗ | ✓ | ✗ |
| Beam Search (Xie et al., 2023) | ✓ | ✗ | ✗ | ✓ | ✗ |
| Reflexion (Shinn et al., 2023) | ✓ | ✓ | ✗ | ✓ | ✓ |
| **LATS (Ours)** | ✓ | ✓ | ✓ | ✓ | ✓ |

Table 1: A summary of related work on reasoning, acting, and planning. LATS is the first work incorporating designs from all three domains, allowing use in all corresponding tasks. We refer to planning as the use of a search algorithm, self-reflection as the use of LM-generated feedback, and external memory as storaging past text context for future updates of solution.

issue by searching over multiple reasoning chains. While these methods enable planning, these methods operate in isolation and do not incorporate external feedback that can improve reasoning.

To help address these issues, we propose LATS (Language Agent Tree Search), a general framework for decision-making and reasoning with language models. LATS unifies LM planning, acting, and reasoning strategies by expanding ReAct (Yao et al., 2023b) into a search over a combinatorial space of possible reasoning and acting steps. We adapt Monte Carlo tree search (MCTS) from model-based reinforcement learning (Silver et al., 2017; Anthony et al., 2017; Jiang et al., 2018) to language agents, repurposing a pretrained LLM as an agent, value function, and optimizer. Utilizing the strong natural language understanding and in-context learning ability of modern LMs, we use text as an interface between each component of the framework, allowing LATS to adapt planning to environmental conditions without additional training. To the best of our knowledge, *LATS is the first framework that combines reasoning, acting, and planning to enhance LLMs*. Notably, LATS doubles the performance of GPT-3.5 on HotPotQA (Yang et al., 2018) over ReAct (Yao et al., 2023b) and raises the average score by 22.1 on WebShop (Yao et al., 2022). When used with GPT-4, LATS achieves a 94.4 Pass@1 rate for programming on HumanEval (Chen et al., 2021), setting the state of the art. To summarize, our **contributions** are the following:

- We introduce an LM-based Monte Carlo tree search variant to deliberately construct the best trajectory from sampled actions, enabling more flexible and adaptive problem-solving compared to reflexive prompting methods. This is guided by heuristics from the LM.

- By integrating external feedback and self-reflection, LATS enhances model sensibility and enables agents to learn from experience, surpassing reasoning-based search methods.

- Through experiments across diverse domains like programming, interactive QA, and web navigation, we demonstrate the versatility of LATS in harnessing LLMs for autonomous reasoning and decision-making.

## 2 RELATED WORK

**LLMs for reasoning.** For LLMs, reasoning typically involves decomposing complex inputs into sequential intermediate steps towards a final answer (Cobbe et al., 2021), demonstrated with Chain-of-Thought (CoT) prompting (Wei et al., 2022) and its variants (Wei et al., 2022; Kojima et al., 2022; Wang et al., 2022). However, these methods, which create chains autoregressively in a single step, often suffer from error propagation as the number of steps increases (Guo et al., 2018; Chen et al., 2022b) due to compound errors. Various advancements aim to mitigate this issue; some approaches, such as Self-Consistency (Wang et al., 2022), employ majority voting over sampled chains, while others focus on multi-step decomposition, such as least-to-most prompting (Zhou et al., 2022), or use of external tools such as a scratchpad (Nye et al., 2021) or compiler (Gao et al., 2022). Recently, CoT has been improved with search algorithms (Yao et al., 2023a; Hao et al., 2023; Besta et al., 2023) that can sample trajectories more effectively. Tree-of-thought (ToT) prompting (Yao et al., 2023a) uses DFS or BFS-based search guided by an LM-generated heuristic while Reasoning via Planning (RAP) (Hao et al., 2023) uses MCTS with rollouts simulated by the LM. However, they rely solely on LM internal knowledge and cannot adapt to useful external feedback.

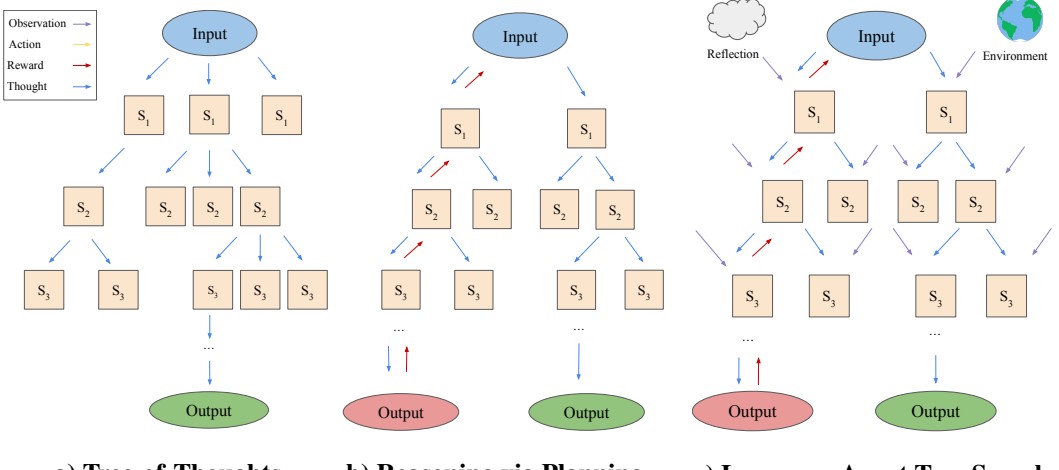

**a) Tree-of-Thoughts**   **b) Reasoning via Planning**   **c) Language Agent Tree Search**

Figure 2: An overview of the differences between LATS and recently proposed LM search algorithms ToT (Yao et al., 2023a) and RAP (Hao et al., 2023). LATS leverages environmental feedback and self-reflection to further adapt search and improve performance.

**LLMs for acting.** The strong reasoning and common-sense abilities of LLMs have also been adapted for decision-making or acting tasks as a policy model in interactive environments. In the realm of robotics LLMs have been employed as high-level controllers of control policies (Ahn et al., 2022; Huang et al., 2022; Driess et al., 2023). Similar work (Baker et al., 2022; Wang et al., 2023; Zhu et al., 2023) has also adapted LLM agents to complex multimodal games such as Minecraft (Guss et al., 2019; Fan et al., 2022). LLMs are particularly useful in text-based environments (Liu et al., 2018; Shridhar et al., 2020; Liu et al., 2023), where acting-based prompting techniques such as ReAct (Yao et al., 2023b) have seen success. Similar to CoT, ReAct is limited by its simplicity and cannot effectively adapt to environment conditions. Many extensions have been proposed to address this, including Self-refine (Madaan et al., 2023) and Reflexion (Shinn et al., 2023; Yao et al., 2023c), which uses self-reflection to enhance reasoning and decision-making, and AdaPlanner (Sun et al., 2023), which incorporates both positive and negative environmental feedback. However these methods focus on refining an individual plan or trajectory and do not consider alternative choices at each step. In addition, recent work (Huang et al., 2023) has suggested LLMs cannot self-correct their internal reasoning, making it critical to use external feedback. Alternatively to pure decision-making environments, the reasoning and practical abilities of LLMs have been enhanced by access to external tools, such as APIs, search engines, calculators, or other models (Schick et al., 2023; Shen et al., 2023; Surís et al., 2023). Contrary to reasoning-based approaches, these methods have not been improved with planning, limiting their effectiveness. We summarize them in Tab. 1.

**Tree-based search.** Tree-based search, where multiple branches of outcomes are explored during search, is widely used in many planning algorithms (Świechowski et al., 2023; LaValle et al., 2001) and Reinforcement Learning (RL) (Hafner et al., 2019; Du et al., 2023; Wu et al., 2023) algorithms for its good exploration-exploitation trade-off. Though tree-based search requires an environment model that can expand from arbitrary state (Vodopivec et al., 2017), which often requires extra training in RL (Hafner et al., 2023), such problem does not exist for LM tasks as we can conveniently backup to any state by setting the input to be the context and corresponding previous output by the LM. Thus, we work on the tree-based framework and use MCTS (Świechowski et al., 2023) to fully release the potential of LMs, while avoiding the cost of training a value function over language descriptions by leveraging the in-context learning (Brown et al., 2020) abilities of LLMs.

## 3 PRELIMINARIES

### 3.1 PROBLEM SETTING AND PROMPTING

Before describing LATS, we first define our problem and outline a few established methods that leverage large language models for reasoning or decision-making. In LM reasoning or decision mak-

ing, we are given an input $x$ in natural language and a pretrained language model $p_\theta(x)$ parameterized by $\theta$; our goal is to generate a final output $y \sim p_\theta(x)$ corresponding to the answer (reasoning) or completes the task (decision-making). Both $x$ and $y$ are language *sequences*, which are comprised of a list of *tokens* (the basic elements of natural language, often words), denoted as $x = (x[1], \ldots, x[n])$ and $y = (y[1], \ldots, y[n])$. The LM decodes text autoregressively, i.e., without other inputs, the probability for an LM to generate a sequence $x$ is given by $p_\theta(x) = \prod_{i=1}^{n} p_\theta(x[i]|x[1 \ldots i-1])$. Usually, to improve the LM, *prompts* are provided along with the input $x$, which are specific instructions or few-shot input-output examples. We denote the generic process where an input $x$ is transformed into an output $y$ by LM: $y \sim p_\theta(y|\texttt{prompt}_{IO}(x))$, where $\texttt{prompt}_{IO}(x)$ denotes the input $x$.

**Chain-of-thought (CoT) Prompting** (Wei et al., 2022) was introduced to cater to scenarios where direct mapping from $x$ to $y$ is intricate, such as when $x$ is from a mathematical query or challenging question. This method hinges on creating *thoughts* $z_1, \ldots, z_n$ that act as stepping stones between $x$ and $y$; each thought $z_i$ is a language sequence. To employ CoT prompting, thoughts are extracted sequentially as $z_i \sim p_\theta^{CoT}(z_i|x, z_{1 \cdots i-1})$, with the final output being $y \sim p_\theta^{CoT}(y|x, z_{1 \cdots n})$.

**Tree-of-thought (ToT) Prompting** (Yao et al., 2023a) extends CoT prompting by exploring multiple reasoning paths over thoughts. It frames problems as a search over a tree where each node $s = [x, z_{1 \cdot i}]$ represents a partial solution state comprising the original input $x$ and thought sequence $z_{1 \cdots i}$. Thoughts $z_i$ are generated by proposal or sampling with CoT $z_i \sim p_\theta^{CoT}(z_i|x, z_{1 \cdots i-1})$. Deliberate search algorithms like breadth-first or depth-first search are used to systematically explore the tree, guided by heuristics based on language model evaluations $V(s)$ of each state.

**Reasoning via Planning** (RAP) (Hao et al., 2023) is similar to ToT, except that MCTS is used over DFS or BFS. Heuristics are designed from an LM, such as the likelihood or confidence of an action, and the LM is used as a world model to predict subsequent states during the simulation step.

**ReAct** (Yao et al., 2023b) extends language models to tasks where the mapping from $x$ to $y$ is enhanced by or requires interactions with an external environment, such as a game or API. This technique constructs an action space $\hat{A} = A \cup Z$ that adds permissible actions $a$ to the reasoning traces $z$ from CoT. Observations $o$ from the environment are used to improve both reasoning and acting. To solve problems with ReAct, after each observation, actions are generated from $p_\theta$ sequentially as $a_i \sim p_\theta^{ReAct}(a_i|x, o_{1 \cdots i-1}, a_{1 \cdots i-1})$, with the final output being $y \sim p_\theta^{ReAct}(y \mid x, o_{1 \cdots n}, a_{1 \cdots n})$.

While the previously described prompting techniques improve LM performance on reasoning tasks, they falter on difficult tasks that involve multifaceted decision-making due to several shortcomings: 1) *Flexibility*: Base prompting methods (CoT or ReAct) autoregressively sample from the LM, neglecting potential alternative continuations from specific states. 2) *Sensibility*: Reasoning-based methods (CoT, RAP, or ToT) rely solely on the internal representations of the LM and cannot consider external observations. This dependency risks fact hallucination and error propagation while setting a performance ceiling. 3) *Adaptability*: Current planning frameworks (RAP or ToT) use simple search algorithms such as BFS or cannot leverage environmental feedback to improve planning. Additionally, the agent is static and cannot reuse previous experience or learn from trial and error. While RAP also adopts MCTS, it is constrained to tasks where the LM can become a world model and accurately predict states. These shortcomings limit the ability of LMs to be deployed as general problem-solving agents and form the motivation for LATS.

## 3.2 MONTE-CARLO TREE SEARCH (MCTS)

Monte-Carlo Tree Search (MCTS) is a heuristic search algorithm that is proved successful on many decision-making environments such as Atari (Ye et al., 2021) and Go (Silver et al., 2016). MCTS builds a decision tree where every node in the tree is a state and edge is an action. MCTS runs for $k$ episodes; for each episode, it starts from the root (i.e., initial state) and iteratively conducts two steps to expand the tree: 1) *Expansion*, where multiple children states $s$ are explored from the current parent state $p$ by sampling $n$ actions, and 2) *Selection*, where the children with the highest UCT *(Upper Confidence bounds applied to Trees)* (Kocsis & Szepesvári, 2006) value is selected by the next iteration. The UCT of a child state $s$ is calculated as follows:

$$UCT(s) = V(s) + w\sqrt{\frac{\ln N(p)}{N(s)}}, \qquad (1)$$

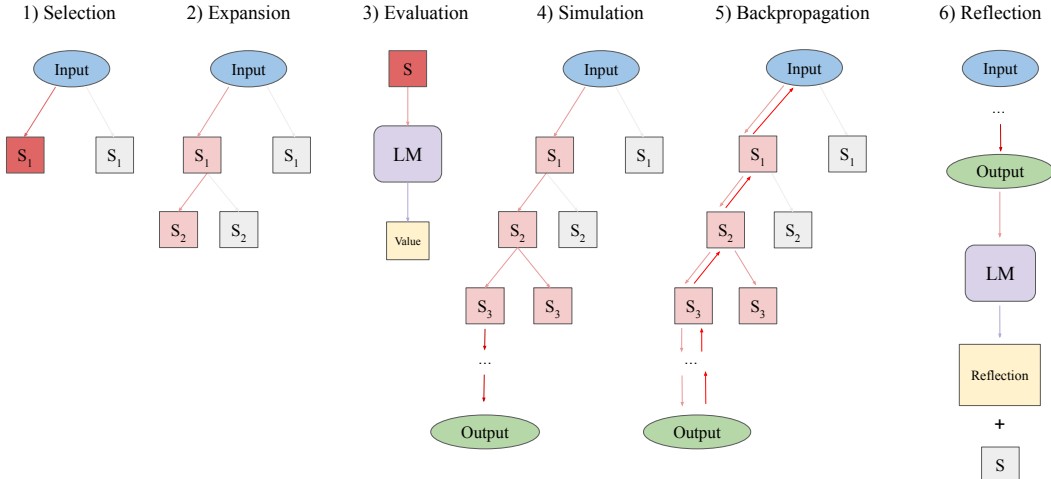

Figure 3: An overview of the six operations of LATS. A node is *selected*, *expanded*, *evaluated*, then *simulated* until a terminal node is reached, then the resulting value is *backpropagated*. If the trajectory fails, a *reflection* is generated and used as additional context for future trials. These operations are performed in succession until the budget is reached or task is successful.

where $N(s)$ is the number of visits to a node $s$, $V(s)$ is the value function (expected return) from the subtree of $s$, $w$ is the exploration weight, and $p$ is the parent node of $s$. The child node with the highest UCT value is selected for expansion in the next iteration. When the end of an episode is reached, a *backpropagation* is carried out: the return $r$ is used for updating every $V(s)$ along the path with the formula $V(s) = \frac{V_{old}(s)(N(s)-1)+r}{N(s)}$, where $V_{old}(s)$ is the old value function. Normally, the major shortcoming of MCTS is that it requires an environment model to undo previous steps and form a searching tree, which is often a strong assumption. However, such a limitation does not exist for LMs, as we can conveniently reset to any step by simply copy-pasting historical text input. Such a special property is the key motivation of our work.

## 4 UNIFYING PLANNING, REASONING, AND ACTING

### 4.1 LM AGENT

LATS supports sequential reasoning or decision-making tasks on the basis of ReAct. At time step $t$, an agent receives an observation $o_t \in O$ from the environment and takes an action $a_t \in A$ following some policy $\pi(a_t|x, o_{1...i-1}, a_{1...i-1})$, where $x$ consists of the task instruction and a number of few-shot examples. We initialize the agent with $p_\theta$ to leverage the useful language representations of an LM as a base decision-maker. We follow the ReAct instantiation in which the action space $\hat{A} = A \cup Z$ consists of both the space of permissible actions $A$ and language space of reasoning traces $Z$. Actions directly affect the environment and result in observation, while thoughts are used to formalize decisions by organizing information, planning future actions, or injecting internal knowledge. The exact instantiation of the action space depends on the particular environment; for decision-making tasks actions might consist of commands on a website while for reasoning tasks the action space might be limited to a few external tools or APIs.

Instead of greedily decoding one trajectory or solution, we sample $n$ actions from $p_\theta$ using the current state. This is based on the intuition that for complex decision-making tasks, there is likely to be a range of potential trajectories or reasoning paths that are correct (Evans, 2010). Sampling a diverse set of candidates at each step mitigates the stochastic nature of LM text generation and enables greater exploration in both the decision-making and reasoning space. We wrap $p_\theta$ within our proposed search algorithm to deliberately construct the best trajectory from sampled actions.

## 4.2 LATS

The main component of LATS is a search algorithm that controls the overall problem-solving process with deliberate planning. To find the most promising trajectory and systemically balance exploration with exploitation, we adopt a variant of Monte Carlo Tree Search (MCTS) that frames decision-making as a tree search, in which each node $s = [x, a_{1...i}, o_{1...i}]$ represents a state comprising the original input $x$, action sequence $a_{1\cdot i}$, and observation sequence $o_{1\cdot i}$.

To adapt MCTS for language agents, LATS repurposes $p_\theta$ as an agent, state evaluator, and feedback generator, leveraging the useful language priors of modern LMs to facilitate planning. While standard MCTS and RAP Hao et al. (2023) rely on internal dynamics models to facilitate simulation, LATS is model-free and uses environment interaction. LATS consists of a series of operations, *selection, expansion, evaluation, simulation, backpropagation, and reflection*, performed in succession until the task is successfully completed or a computational limit is reached. The full psuedocode of LATS can be found in Sec. A in the Appendix.

**Selection.** In the first operation, the algorithm identifies a segment of the current tree most suitable for subsequent expansion. Starting from the root node, denoted as the initial state $s_0$, a child node is selected at each tree level until a leaf node is reached. To balance exploration and exploitation, we use the UCT algorithm as shown in Eq. 1.

**Expansion.** After selecting a node, the second operation expands the tree by sampling $n$ actions from $p_\theta$, as described in the prior section. The environment receives each action and returns corresponding feedback as an observation. This results in $n$ new child nodes added to the tree. This tree is stored in an external long-term memory structure.

**Evaluation.** The third operation assigns a scalar value to each new child node to be used for selection and backpropagation. This value effectively quantifies the agent's progress in task completion, serving as a heuristic to steer the search algorithm towards the most promising regions of the tree. Following Yao et al. (2023a) we repurpose $p_\theta$ into a value function by prompting it to reason about a given state. To obtain a scalar value, we instruct $p_\theta$ to end its reasoning trace with a score indicating the correctness of the trajectory. This method offers enhanced flexibility over programmed heuristics (Campbell et al., 2002) and greater efficiency than learned heuristics (Silver et al., 2017).

**Simulation.** The fourth operation expands the currently selected node until a terminal state is reached. At each depth level we sample and evaluate nodes with the same operations, but prioritize nodes of highest value. Reaching a terminal state provides objective feedback on the correctness of a trajectory. If the task is completed successfully, then LATS terminates the search. If the solution is partially successful or unsuccessful, then we perform two additional operations as described below.

**Backpropagation.** This operation updates the values of the tree based on the outcome of a trajectory. For each node $s_0, s_1, \ldots, s_n$ in the trajectory from root (initial state $s_0$) of the searching tree to leaf (terminal state $s_n$), its value is updated to reflect the outcome of the simulation by $N(s_i) = N_{\text{old}}(s_i) + 1$ and $V(s_i) = \frac{r + N_{\text{old}}(s_i) V_{\text{old}}(s_i)}{N(s_i)}$, where $r$ is the return and $N_{\text{old}}, V_{\text{old}}$ are the old number of visits and value function. These updated values are used in the UCT formula (Eq. 1) to guide the selection of the next node for exploration.

**Reflection.** In addition to the environmental feedback, we also leverage *self-reflection* to further refine the decision-making process (Shinn et al., 2023; Madaan et al., 2023). Upon encountering an unsuccessful terminal node, $p_\theta$ is prompted with the trajectory and final reward to provide a verbal self-reflection that summarizes the errors in the reasoning or acting process and proposes superior alternatives. We store both failed trajectories and corresponding reflections in the memory. In subsequent iterations, these are integrated as additional context to the agent and value function, refining both through in-context learning. This imparts a semantic gradient signal more useful than a scalar value, enabling the agent to learn from trial and error without the cost of expensive optimization processes such as reinforcement learning.

Conceptually, LATS has the following advantages as a general framework for reasoning and decision-making with LM agents. (1) *Generality*: LATS supports both reasoning and decision-making tasks by defining a shared space of thoughts and actions. (2) *Deliberate*: The use of MCTS and LM value function ensures a principled search that selects options with high value while explor-

| Prompt Method | HotpotQA (EM) |
|---|---|
| I/O | 0.32 |
| CoT (Wei et al., 2022) | 0.34 |
| CoT - SC (Wang et al., 2022) | 0.38 |
| ToT (Yao et al., 2023a) | 0.55 |
| RAP (Hao et al., 2023) | 0.60 |
| RAP (n = 10) | 0.60 |
| LATS (CoT) | **0.60** |

| Prompt Method | HotpotQA (EM) |
|---|---|
| ReAct (Yao et al., 2023b) | 0.32 |
| ReAct (best of k) | 0.38 |
| Reflexion (Shinn et al., 2023) | 0.51 |
| LATS | 0.61 |
| LATS (n = 1) | 0.48 |
| LATS (n = 3) | 0.56 |
| LATS (n = 10) | 0.64 |
| LATS (CoT + ReAct) | **0.71** |

Table 2: GPT-3.5 reasoning-based prompting (left) and acting-based prompting (right) results on HotpotQA. LATS achieves the highest exact match (EM) for acting and is competitive on reasoning. Unless otherwise specified, we sample $n = 5$ nodes during expansion and $k = 50$ trajectories.

ing promising alternatives. (3) *Adaptability*: LATS is designed around the use of external feedback through observations and self-reflection, enabling greater adaptation during problem-solving. (4) *Flexibility*: LATS can accommodate different scenarios, environments, and resource stipulations by modifying state design and tree dimensions. (5) *Modularity*: The base LM agent, reflection generator, and value function can be independently altered and adapted to individual LM properties.

## 5 EXPERIMENTS

To demonstrate the general applicability of LATS, we evaluate our method on a variety of decision-making domains that requires both reasoning and acting ability: programming (Chen et al., 2021; Austin et al., 2021), HotPotQA (Yang et al., 2018), and WebShop (Yao et al., 2022).

### 5.1 HOTPOTQA

For a task that can be approached with both reasoning-based and acting-based strategies, we consider HotPotQA (Yang et al., 2018), a multi-hop question-answering benchmark that requires retrieval over two or more Wikipedia passages. For the action space, in addition to LM thoughts we follow the setup from Yao et al. (2023b), which provides the agent with API calls to search and lookup information. The output of these API calls and self-generated reflections form the observation space. We use a subset of 100 questions and three few-shot examples for each method. For ToT, we use DFS as the base search algorithm and scoring with the LM as the heuristic. For all methods that involve sampling, including LATS, we sample $k = 50$ trajectories. More details and prompts can be found in Sec. D and Sec. E in the Appendix.

We evaluate internal reasoning strategies by removing actions and observations from the context, corresponding to CoT (Wei et al., 2022) and its variants, CoT-SC (Wang et al., 2022), ToT (Yao et al., 2023a), and RAP (Hao et al., 2023). These methods rely solely on the agent's existing knowledge to answer the question. We also consider acting-based methods ReAct, Reflexion, and LATS, which augment the agent with the interactive API environment and primarily evaluate its information retrieval abilities. While LATS is designed for scenarios where external feedback can enhance reasoning, we also implement a reasoning-only version with CoT as the base prompt. We also combine internal and external reasoning in LATS by first prompting with a CoT-based prompt, then switching to a ReAct-based prompt upon failure. This is closer to how humans might approach this task, by using tools to lookup additional information only when the answer is not already known.

**Results.** We observe in Tab. 2 that both internal reasoning and external retrieval strategies perform well on HotPotQA. Due to their large-scale training corpus, modern LLMs already encode factual knowledge and can often directly answer the question correctly. While CoT can slightly enhance performance on questions requiring reasoning, larger gains are observed with search methods ToT and RAP, which can sample and explore more outputs. We observe similar results for acting-based methods. LATS surpasses ReAct, even when sampling the same number of trajectories, by expanding more nodes with principled search (see Fig. 5 in Appendix D for a qualitative sample). This is demonstrated when modifying $n$, the number of nodes expanded during each iteration. Increasing $n$

| Prompt Method | Model | Pass@1 |
|---|---|---|
| CoT (Wei et al., 2022) | GPT-3.5 | 46.9 |
| ReAct (Yao et al., 2023b) | GPT-3.5 | 56.9 |
| Reflexion (Shinn et al., 2023) | GPT-3.5 | 68.1 |
| ToT (Yao et al., 2023a) | GPT-3.5 | 54.4 |
| RAP (Hao et al., 2023) | GPT-3.5 | 63.1 |
| LATS (Ours) | GPT-3.5 | **86.9** |
| I/O | GPT-4 | 80.1 |
| Reflexion | GPT-4 | 91.0 |
| LATS | GPT-4 | **94.4** |

| Prompt Method | Pass@1 |
|---|---|
| CoT (Wei et al., 2022) | 54.9 |
| ReAct (Wei et al., 2022) | 67.0 |
| Reflexion (Shinn et al., 2023) | 70.0 |
| ToT (Yao et al., 2023a) | 65.8 |
| RAP (Hao et al., 2023) | 71.4 |
| LATS (Ours) | **81.1** |

Table 3: GPT-3.5 and GPT-4 Pass@1 accuracy on HumanEval (Chen et al., 2021) and MBPP (Austin et al., 2021). Prompting with LATS achieves the highest performance. We sample 5 solutions during expansion for 8 iterations.

can consistently improve performance, although at greater computational and inference costs. LATS is also competitive to RAP on internal reasoning but performs worse than acting. Combining internal and external reasoning in LATS results in the highest performance, indicating the importance of external feedback in augmenting reasoning even in tasks the base LM can already perform.

## 5.2 PROGRAMMING

To demonstrate the importance of external observations for complex reasoning tasks, we evaluate the baselines and LATS on programming with Humaneval (Chen et al., 2021) and MBPP (Austin et al., 2021). Both datasets measure the correctness of synthesized programs in Python from natural language docstrings. We use individual solutions as the action space and test suite and compiler feedback as the external observation. We follow Chen et al. (2022a) and use an LLM to generate a synthetic test suite of syntactically valid "assert" statements for each question. For each step, the solution is evaluated on this test suite, and the results including successful and failed tests and compiler output, are added to the context as an observation. We use the same test suite for Reflexion.

For this task, the reasoning and acting baselines share an action space, but acting methods are able to incorporate observations as additional context. For LATS, since each action corresponds to a complete solution, we skip the simulation step of LATS and directly use the percentage of passed tests as the backpropagated reward. We use $k = 8$ iterations, set the number of generated tests at 4, and sample $n = 5$ solutions during expansion. After the search is completed, we select the solution with the highest value and evaluate it on the real test suite for the pass@1 accuracy evaluation. More details and prompts can be found in Sec. D and Sec. F in the Appendix.

**Results.** We find in Tab 3 that both search and semantic feedback are crucial for better performance. Despite not using observations, ToT and RAP are competitive with Reflexion. LATS has the highest performance on both datasets. Since RAP uses a similar search algorithm as LATS, this reveals the importance of external feedback for difficult reasoning tasks such as programming. With GPT-4, using LATS sets the state of the art for HumanEval, showing LATS can be used with more advanced LLMs for higher performance.

## 5.3 WEBSHOP

For a complex decision-making environment with practical applications, we consider WebShop (Yao et al., 2022), an online shopping environment composed of a website with 1.18M real-world products and 12k human instructions. Agents must navigate a website through a variety of commands to purchase an item matching a user specification. We use the preconstructed action space of search and click commands and browser feedback and reflections for the observation. The performance is gauged using two metrics: an average score, reflecting the percentage of user-specified attributes met by the selected product, and a success rate, indicating the frequency with which the chosen product fulfills all given conditions. We compare against acting-based prompting methods and RL-based approaches. We evaluate on 50 instructions, expand $n = 5$ children for LATS, and set $k = 30$ for LATS, ReAct best of $k$, and Reflexion. More details and prompts are in Appendix D and G.

| Method | Score | SR |
|---|---|---|
| ReAct (Yao et al., 2023b) | 53.8 | 28.0 |
| ReAct (best of k) | 59.1 | 32.0 |
| Reflexion (Shinn et al., 2023) | 64.2 | 35.0 |
| LATS | **75.9** | **38.0** |
| IL Yao et al. (2022) | 59.9 | 29.1 |
| IL+RL Yao et al. (2022) | 62.4 | 28.7 |
| Fine-tuning (Furuta et al., 2023) | 67.5 | 45.0 |
| Expert | 82.1 | 59.6 |

Table 4: Score and success rate (SR) on Webshop, separated into prompting, training, and human performance. For the same number of iterations, LATS improves both score and success rate and surpasses RL-based training.

| Prompt Method | HotPotQA |
|---|---|
| ReAct (best $k = 250$) | 0.42 |
| CoT-SC ($k = 250$) | 0.40 |
| ToT (ReAct) | 0.49 |
| RAP (ReAct) | 0.54 |
| LATS (DFS) | 0.53 |
| LATS (No Reflection) | 0.56 |
| LATS | 0.61 |

Table 5: Standardized comparisons between LATS, ToT, and RAP; we use ReAct as the base prompt for all methods and sample $n = 5$ children for MCTS and DFS and $k = 50$ maximum trajectories.

**Results.** We find in Tab. 4 that GPT-3.5 with ReAct is competitive to imitation learning, and can exceed reinforcement learning techniques with stronger prompting strategies. Sampling $k = 30$ trajectories with ReAct and Reflexion results in a similar performance, suggesting the semantic feedback is not as helpful in complex environments like WebShop. Indeed like in Shinn et al. (2023), we find that generated reflections are often generic and do not provide useful feedback, resulting in a tendency for the agent to become stuck in local minima. However, using LATS indeed results in a noticeable improvement, indicating a more effective exploration for the same number of iterations.

### 5.4 Additional Observations

We also conduct additional experiments on HotPotQA to demonstrate the effect of each component of LATS. We also design a version of ToT and RAP with ReAct prompt and can handle external observations. We use HotPotQA as our setup incorporates both reasoning (through thoughts) and acting (through API calls); the results are shown in Tab. 5. More ablations for token consumption on HotPotQA are in Tab. 7 in Appendix C. Note that baselines generally perform worse than the reasoning-only setting of HotPotQA, which indicates that the acting-based setting is more challenging and adaption of search algorithms to decision-making scenarios is non-trivial.

**Self-reflection.** We use self-reflection to provide additional semantic signals for the agent. We observe a 0.05 performance drop when removed from LATS, suggesting this is useful. This is a smaller gain Reflexion (Shinn et al., 2023) observes over ReAct (Yao et al., 2023b) as shown in Tab. 2, suggesting overlap between the types of questions where there is an improvement with self-reflection and search. This variant outperforms RAP-ReAct, reflecting our improvements to MCTS.

**Search Algorithm.** MCTS is a more principled search algorithm than variants like A* or DFS search and the basis for observed performance gains. We observe the effects of using DFS, and incorporate the LM-based heuristic used in ToT (Yao et al., 2023a) in which branches with low values are pruned. This removes the selection and backpropagation operations, and we observe a 0.08 drop in performance when sampling the same number of nodes, but outperforms ToT-ReAct.

### 6 Conclusion

In this work, we introduce Language Agent Tree Search (LATS), the first framework to unify planning, acting, and reasoning for enhanced LLM problem solving. By deliberately constructing trajectories with search algorithms, incorporating external feedback, and enabling agents to learn from experience, LATS addresses key limitations of prior prompting techniques. Our evaluations demonstrate the ability of LATS to harness LLM capabilities for a variety of decision-making tasks while keeping its reasoning ability without additional training. The proposed synergies between search, interaction, and reflection offer a versatile approach to autonomous decision-making, highlighting the potential of LLMs as generalist agents. A full discussion of the limitations and broader impacts is in Appendix B.

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

## 7 APPENDIX

The appendix is organized as follows. First in Sec. A, we show the pseudocode of our proposed algorithm, LATS; then in Sec. B, we provide further discussion of our method and its limitations, future direction and broader impact; then in Sec. C we provide additional experimental results; then in Sec. D, we specify the environment details in our experiments; finally, we list our prompts used for the three environments in Sec. E (HotPotQA), Sec. F (Programming) and Sec. G (Webshop) respectively.

## A LATS PSEUDOCODE

Alg. 1 shows the pseudocode of our algorithm LATS. Nodes are stored explicitly in the memory. Unless otherwise specified, in all experiments we use $n = 5$ and $w = 1$.

---

**Algorithm 1** $\text{LATS}(S_0, p_\theta, p_V, p_{\text{ref}}, d, k, n, w)$

---

**Require:** Initial state $s_1$, action generator $p_\theta$, value function $p_V$, reflection generator $p_{\text{ref}}$, number of generated actions $n$, depth limit $L$, number of roll-outs $K$, context $c$, and exploration weight $w$

Initialize action space $A$, observation space $O$

Initialize the state-action value function $p_V : S \times A \mapsto \mathbb{R}$ and visit counter $N : S \mapsto \mathbb{N}$ to zero

**for** $k \leftarrow 0, \ldots, K - 1$ **do**
    **for** $t \leftarrow 0, \ldots, L - 1$ **do**
        **if** $s_t$ not terminal **then**                            ▷ Expansion & Simulation
            **for** $i \leftarrow 1, \ldots, n$ **do**
                Sample $a_t^{(i)} \sim p_\theta(a \mid s_t)$
                Get $o_t^{(i)}$ from environment, $s_{t+1}^{(i)} \leftarrow (c_t^{(i)}, o_t^{(i)}, a_t^{(i)}), c_{t+1}^{(i)} \leftarrow (o_t^{(i)}, a_t^{(i)})$
                Evaluate $V_t^{(i)} \sim p_V(s_t^{(i)})$                      ▷ Evaluation
                $V(s_t) \leftarrow V_t^{(i)}$
                Add $s_t^{(i)}$ to children
            **end for**
        **end if**
        **if** $s_t$ is terminal **then**                                ▷ Reflection
            Get $r$ from environment
            **if** $r$ not success **then**
                reflection $\leftarrow p_{\text{ref}}(c_t)$
                $c \leftarrow$ reflection
            **end if**
        **end if**
        $a_t \leftarrow \arg\max_{a \in e(s_t)} \left[ V(s_t) + w\sqrt{\frac{\ln N(s_{t-1})}{N(s_t)}} \right]$              ▷ Selection
        $N(s_{t+1}) \leftarrow N(s_{t+1}) + 1$
        **if** $a_t$ is an output action **then break**
    **end for**
    $T \leftarrow$ the actual number of steps
    **for** $t \leftarrow T - 1, \ldots, 0$ **do**                                ▷ Backpropagation
        $V(s_t) \leftarrow \frac{V(s_t)(N(s_t)-1)+r}{N(s_t)}$
    **end for**
**end for**

---

## B DISCUSSION

**Limitations.** Although LATS can improve reasoning and decision-making, this arrives at a higher computational cost relative to simpler prompting methods like ReAct or Reflexion. The search process takes more time than standard prompting or simpler techniques, and requires greater inference costs. While such an issue is mitigated by the fact that the number of nodes $n$ expanded at every step provides a natural trade-off between performance and efficiency (setting $n = 1$ makes the method as effecient as ReAct with multiple trials or CoT-SC), in practice we recommend using LATS for difficult tasks like programming or for situations where performance is prioritized over efficiency.

| Prompt Method | HotpotQA (EM) |
|---|---|
| LATS (w=0.5) | 0.55 |
| LATS (w=2.0) | 0.61 |
| LATS (d=4) | 0.58 |
| LATS (CoT) | 0.60 |
| LATS (No LM Heuristic) | 0.37 |
| LATS | 0.61 |

Table 6: Ablation results on LATS and baseline variants in HotPotQA measured by Exact Match (EM). We test different depth $d$, exploration factor $w$, and versions of LATS using CoT and without the LM value function. We sample $n = 5$ and $k = 50$ trajectories.

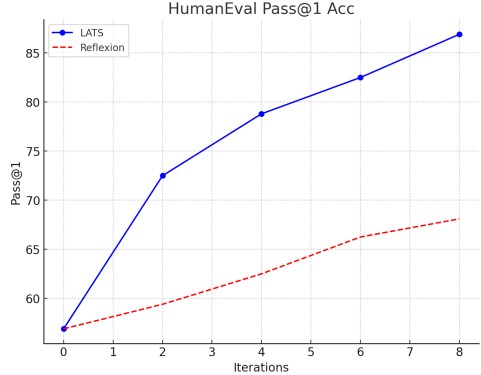

Figure 4: Performance over successive iterations on HumanEval with GPT-3.5.

We hope that continued advancements in LLMs will reduce costs and increase the practicality of LATS.

Additionally, the benchmarks we use in this paper are relatively simple and focused on decision-making, compared to the complexity of real-world interactive environments. In addition, some environments might not easily support rollbacks to previous states. However, the design of LATS is flexible and can be adjusted to various resource constraints. Using planning-based prompting methods like LATS in environments like Minecraft (Fan et al., 2022) and more reasoning benchmarks would be interesting avenues for future work.

**Broader impact.** LATS is a framework that enhances LLM performance through interactions with an environment. This improvement in autonomous decision-making may facilitate harmful uses of LLMs. Alternatively, LATS enhances interpretability and the potential for greater alignment, as it generates understandable, high-level linguistic reasoning and actions through several rounds of decision-making and reflection, rather than relying on implicit, low-level token values.

## C  ABLATIONS

In this section, we ablate various designs of LATS. Experiments are conducted on HotPotQA with a maximum of $k = 50$ trajectories and sampling size of $n = 5$ and HumanEval with a maximum of $k = 8$ trajectories and sampling size of $n = 5$. The result for HotPotQA is shown in Tab. 5 and HumanEval in Fig. 4.

**Exploration weight.** We find that there is lower performance on HotPotQA when the exploration weight $w$ in the selection formula is decreased to $0.5$, suggesting that this reduces the effectiveness of the search. Increasing $w$ to $2.0$ does not lead to a performance improvement, but we tend to observe faster convergence. The optimal setting depends on the particular environment and complexity of the state space.

**Depth.** In our main experiments we use a maximum depth of $d = 7$ on HotPotQA for all methods, following previous work (Yao et al., 2023b). We ablate the effect on LATS after reducing it to $d = 4$. This results in only a slight drop in performance. We find that most questions can be answered within four steps, and using a greater number of steps tends to force the agent into local minima and rarely improves success.

**LM value function.** The LM value function scores states based on expected future reward. Without this heuristic, the only signal to guide search would be from environment rewards for completed trajectories, which are scarce and often binary. When we remove the evaluation operation, we observe a dramatic $0.24$ drop in performance.

**Performance over time.** To see the effects of increasing the number of trajectories sampled, we change $k$ to different values. We conduct this experiment on HumanEval, which has a more notice-

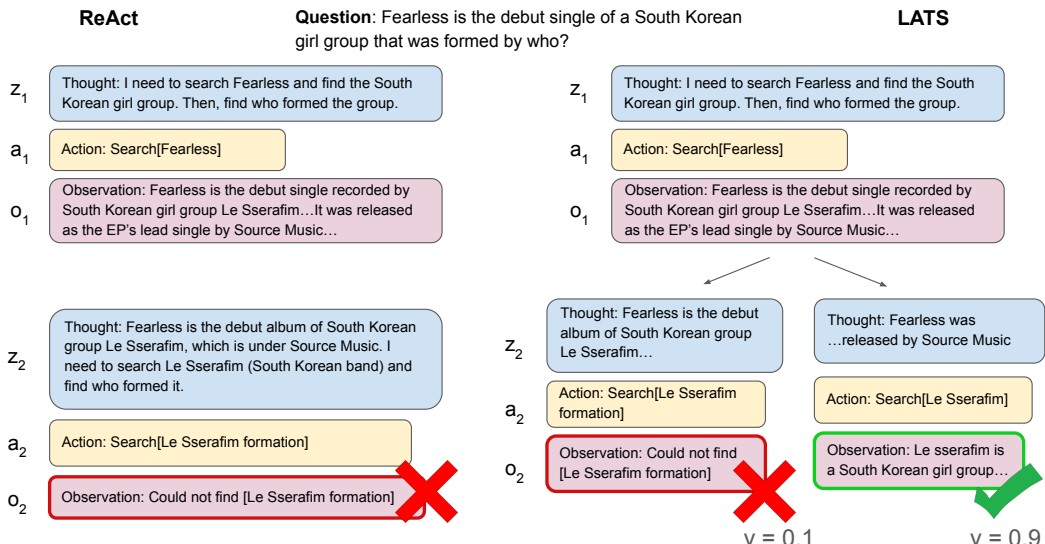

Figure 5: Example trajectories on HotPotQA for ReAct (left) and LATS (right). LATS can sample more actions and avoid failure from previous mistakes by evaluating states with an LM to guide the search toward promising areas of the tree.

able difference due to sampling less trajectories. The results are shown in Fig. 4, in which LATS scales better with more iterations than Reflexion.

**Sample complexity and Token cost.** One possible concern of LATS is that the tree-structured search might consume much more tokens than existing methods. To further study the computational cost of LATS compared to prior methods, we examine the sample complexity (i.e. asymptotic token cost) of all methods considered in this paper, and count the average number of nodes expanded by our method and other tree-structured methods (ToT and RAP) upon successful search on HotPotQA. We present the results in Tab. 7; the result shows that our method has the same sample complexity as other tree-based search methods, and has less average number of nodes expanded upon success, which indicates less token cost. The token cost gap will be even larger when taking failed trajectories into account, since our method has higher success rate and reaches computational budget limit less often.

| Method | Performance (↑) | Sample complexity (↓) | Avg. #nodes upon success (↓) | Nodes expanded |
|---|---|---|---|---|
| ReAct (Best $k = 250$) | 0.42 | $O(k)$ | N/A | |
| CoT-SC ($n = 1, k = 250$) | 0.40 | $O(k)$ | N/A | |
| LATS ($n = 1, k = 50$) | 0.48 | $O(k)$ | N/A | |
| ToT (ReAct) | 0.49 | $O(kn)$ | 84.05 | 210215 |
| RAP (ReAct) | 0.54 | $O(kn)$ | 70.60 | 176500 |
| LATS ($n = 5, k = 50$) | 0.61 | $O(kn)$ | 66.65 | 173290 |

Table 7: The performance, sample complexity of different methods and average number of nodes expanded upon success by methods with tree-based search. $n$ is the number of children nodes expanded at every step and $k$ is the number of trajectories. Our method has the same sample complexity as other methods with tree-based search and expands less nodes upon success, which indicates lower token cost.

## D  ENVIRONMENT DETAILS

### D.1  HOTPOTQA

HotPotQA (Yang et al., 2018) is a question-answering dataset that requires reasoning over multiple supporting documents to answer questions. It contains 113k Wikipedia-based question-answer

pairs crafted by crowdworkers to be diverse, multi-hop, and explainable. Questions cover a range of types like entities, locations, dates, and comparison of shared properties between two entities. Crowdworkers also provide supporting facts from the documents that justify the answer. We use the HotPotQA benchmark setting with all the Wikipedia paragraphs to test retrieval. We use a randomly selected subset of 100 questions for our experiments and a maximum depth limit of 6. Fig. 5 illustrates how ReAct and LATS work on an example task of HotPotQA, and gives a qualitative example on how LATS outperforms ReAct on the task.

**Action Space.** We adopt the Wikipedia web API proposed in Yao et al. (2023b), with three types of actions to support interactive information retrieval:

(1) **search**[entity], which returns the first 5 sentences from the corresponding entity wiki page if it exists, or else suggests top-5 similar entities from the Wikipedia search engine,

(2) **lookup**[string], which returns the next sentence in the page containing string,

(3) **finish**[answer], which finishes the current task with answer.

These API calls and free-form thoughts form the action space for this environment.

### D.2 PROGRAMMING

The HumanEval dataset (Chen et al., 2021) is a collection of 164 handwritten programming problems introduced to evaluate the functional correctness of models for synthesizing programs from natural language descriptions. Each problem includes a function signature, docstring description, reference implementation, and multiple unit tests, with an average of 7.7 tests per problem. The programming tasks assess comprehension of natural language, reasoning, algorithms, and basic mathematics, at a difficulty level comparable to simple software interview questions. Pass rates are evaluated with the pass@k metric, where k samples are generated per problem and a problem is considered solved if any sample passes all tests. We use all 164 problems for our experiments and a maximum depth limit of 8.

The Mostly Basic Programming Problems (MBPP) Austin et al. (2021) benchmark contains 974 short Python functions designed to evaluate program synthesis techniques. The dataset was constructed by crowdsourcing from workers with basic Python knowledge. Each data point consists of a natural language description of a programming task, a reference solution implementation, and three test cases for functional correctness. The natural language prompts are typically short, one-sentence descriptions. Solutions cover common programming constructs including mathematical operations, list processing, string manipulation, and usage of the Python standard library. On average, solutions are 6.8 lines of code. The dataset is also supplemented with an additional set of 426 problems that were manually verified for unambiguous specifications, standard function signatures, and accurate test cases. We use a randomly selected subset of 397 problems for our experiments.

### D.3 WEBSHOP

WebShop (Yao et al., 2022) is an interactive web-based environment designed to evaluate agents on grounded language understanding and decision-making. It simulates an e-commerce shopping task by providing agents with over 1 million real-world products scraped from Amazon, spanning 5 categories and 113 subcategories. These products contain rich linguistic information, with an average text length of 262 words and a vocabulary size of 224k. In addition, there are over 800k unique product options available for customization. The environment renders webpages in two modes: HTML mode provides pixel-level observations with interactive elements, while simple mode converts the raw HTML into a structured text observation more amenable for training agents. The action space consists of query searches and button clicks, which transition between 4 page types: search, results, item and item-detail. Instructions are crowdsourced natural language specifying product attributes and options, with a total of 12k collected. Automatic rewards are computed by comparing the product purchased by the agent against the attributes and options specified in the instruction, using both lexical matching and semantic similarity metrics.

There are two evaluation metrics used in WebShop: (1) **Task Score**: defined as ($100 \times$ avg. reward), which captures the average reward obtained across episodes; and (2) **Success Rate (SR)** defined as

| Type | Argument | State → Next State |
|------|----------|--------------------|
| search | [*Query*] | Search → Results |
| choose | Back to search | ∗ → Search |
| choose | Prev/Next page | Results → Results |
| choose | [*Product title*] | Results → Item |
| choose | [*Option*] | Item → Item |
| choose | Desc/Overview | Item → Item-Detail |
| choose | Previous | Item-Detail → Item |
| choose | Buy | Item → Episode End |

Table 8: Action space of webshop.

the portion of instructions where $r = 1$. The reward is calculated based on the number of attributes satisfied by the selected item. We use 50 environments for our experiments and a maximum depth limit of 15.

# E HOTPOTQA PROMPTS

## E.1 BASE ACTING PROMPT

Solve a question answering task with interleaving Thought, Action, Observation steps. Thought can reason about the current situation, and Action can be three types:

(1) Search[entity], which searches the exact entity on Wikipedia and returns the first paragraph if it exists. If not, it will return some similar entities to search.

(2) Lookup[keyword], which returns the next sentence containing keyword in the current passage.

(3) Finish[answer], which returns the answer and finishes the task.

After each observation, provide the next Thought and next Action.

Here are some examples:

Question: Which magazine was started first Arthur's Magazine or First for Women?

Thought 1: I need to search Arthur's Magazine and First for Women, and find which was started first.

Action 1: Search[Arthur's Magazine]

Observation 1: Arthur's Magazine (1844ᄋ̆0e2ᄋ̆008ᄋ̆00931846) was an American literary periodical published in Philadelphia in the 19th century. Edited by Timothy Shay Arthur, it featured work by Edgar A. Poe, J.H. Ingraham, Sarah Josepha Hale, Thomas G. Spear, and others.[1][2] In May 1846 it was merged into Godey's Lady's Book.[3]

Thought 2: Arthur's Magazine was started in 1844. I need to search First for Women next.

Action 2: Search[First for Women]

Observation 2: First for Women is a woman's magazine published by Bauer Media Group in the USA.[1] The magazine was started in 1989. It is based in Englewood Cliffs, New Jersey.[2] In 2011 the circulation of the magazine was 1,310,696 copies.[3]

Thought 3: First for Women was started in 1989. 1844 (Arthur's Magazine) ¡ 1989 (First for Women), so Arthur's Magazine was started first.

Action 3: Finish[Arthur's Magazine]

(examples)

You have attempted to answer the following question before and failed. The following reflection(s) give a plan to avoid failing to answer the question in the same way you did previously. Use them to improve your strategy of correctly answering the given question.

(trajectories)

(input)

## E.2 BASE REASONING PROMPT

Solve a question answering task by having a Thought, then Finish with your answer. Thought can reason about the current situation. Finish[answer] returns the answer and finishes the task. You will be given context that you should use to help you answer the question. Start your response with either Action or an indexed Thought

Here are some examples:

Question: What is the elevation range for the area that the eastern sector of the Colorado orogeny extends into?

Let's think step by step.

Thought 1: The eastern sector of Colorado orogeny extends into the High Plains.

Thought 2: High Plains rise in elevation from around 1,800 to 7,000 ft

Thought 3: The answer is 1,800 to 7,000 ft.

Action: Finish[1,800 to 7,000 ft]

(examples)

Previous trial: (trajectories)

(input)

## E.3 VALUE FUNCTION PROMPT

Analyze the trajectories of a solution to a question answering task. The trajectories are labeled by environmental observations about the situation, thoughts that can reason about the current situation and actions that can be three types:

(1) Search[entity], which searches the exact entity on Wikipedia and returns the first paragraph if it exists. If not, it will return some similar entities to search.

(2) Lookup[keyword], which returns the next sentence containing keyword in the current passage.

(3) Finish[answer], which returns the answer and finishes the task.

Given a question and a trajectory, evaluate its correctness and provide your reasoning and analysis in detail. Focus on the latest thought, action, and observation. Incomplete trajectories can be correct if the thoughts and actions so far are correct, even if the answer is not found yet. Do not generate additional thoughts or actions. Then at the last line conclude "Thus the correctness score is s", where s is an integer from 1 to 10.

Question: Which magazine was started first Arthur's Magazine or First for Women?

Thought 1: I need to search Arthur's Magazine and First for Women, and find which was started first.

Action 1: Search[Arthur's Magazine]

Observation 1: Arthur's Magazine (1844Ŏe2Ŏ080Ŏ0931846) was an American literary periodical published in Philadelphia in the 19th century. Edited by Timothy Shay Arthur, it featured work by Edgar A. Poe, J.H. Ingraham, Sarah Josepha Hale, Thomas G. Spear, and others.[1][2] In May 1846 it was merged into Godey's Lady's Book.[3]

This trajectory is correct as it is reasonable to search for the first magazine provided in the question. It is also better to have simple searches corresponding to a single entity, making this the best action.

Thus the correctness score is 10

(other examples)

(failed trajectories)

(context)

### E.4 REFLECTION PROMPT

Analyze the trajectories of a solution to a question answering task. The trajectories are labeled by environmental observations about the situation, thoughts that can reason about the current situation and actions that can be three types:

(1) Search[entity], which searches the exact entity on Wikipedia and returns the first paragraph if it exists. If not, it will return some similar entities to search.

(2) Lookup[keyword], which returns the next sentence containing keyword in the current passage.

(3) Finish[answer], which returns the answer and finishes the task.

Given a question and a trajectory, evaluate its correctness and provide your reasoning and analysis in detail. Focus on the latest thought, action, and observation. Incomplete trajectories can be correct if the thoughts and actions so far are correct, even if the answer is not found yet. Do not generate additional thoughts or actions. Then at the last line conclude "Thus the correctness score is s", where s is an integer from 1 to 10.

Question: Which magazine was started first Arthur's Magazine or First for Women?

Thought 1: I need to search Arthur's Magazine and First for Women, and find which was started first.

Action 1: Search[Arthur's Magazine]

Observation 1: Arthur's Magazine (1844ǒ0e2ǒ080ǒ0931846) was an American literary periodical published in Philadelphia in the 19th century. Edited by Timothy Shay Arthur, it featured work by Edgar A. Poe, J.H. Ingraham, Sarah Josepha Hale, Thomas G. Spear, and others.[1][2] In May 1846 it was merged into Godey's Lady's Book.[3]

This trajectory is correct as it is reasonable to search for the first magazine provided in the question. It is also better to have simple searches corresponding to a single entity, making this the best action.

Thus the correctness score is 10

(other examples)

(failed trajectories)

(context)

## F PROGRAMMING PROMPTS

### F.1 HUMANEVAL FUNCTION IMPLEMENTATION EXAMPLE

Sample function signature:

```
def minSubArraySum(nums):
    Given an array of integers nums, find the minimum sum of any
    non-empty sub-array of nums.
    Example
    minSubArraySum([2, 3, 4, 1, 2, 4]) == 1
    minSubArraySum([-1, -2, -3]) == -6
```

Sample function body implementation:

```
min_sum = float('inf')
for i in range(len(nums)):
    current_sum = 0
    for j in range(i, len(nums)):
        current_sum += nums[j]
        if current_sum < min_sum:
            min_sum = current_sum
return min_sum
```

## F.2 BASE ACTING/REASONING PROMPT

You are an AI Python assistant. You will be given your previous implementation of a function, a series of unit tests results, and your self-reflection on your previous implementation. Write your full implementation (restate the function signature).

Example 1:

[previous impl]:

```
def add(a: int, b: int) -> int:
    """
    Given integers a and b, return the total value of a and b.
    """
    return a - b
```

[unit test results from previous impl]:

Tested passed:

Tests failed:

assert add(1, 2) == 3 # output: -1

assert add(1, 2) == 4 # output: -1

[reflection on previous impl]:

The implementation failed the test cases where the input integers are 1 and 2. The issue arises because the code does not add the two integers together, but instead subtracts the second integer from the first. To fix this issue, we should change the operator from '-' to '+' in the return statement. This will ensure that the function returns the correct output for the given input.

[improved impl]:

```
def add(a: int, b: int) -> int:
    """
    Given integers a and b, return the total value of a and b.
    """
    return a + b
```

## F.3 REFLECTION PROMPT

You are a Python programming assistant. You will be given a function implementation and a series of unit test results. Your goal is to write a few sentences to explain why your implementation is wrong as indicated by the tests. You will need this as guidance when you try again later. Only provide the few sentence description in your answer, not the implementation. You will be given a few examples by the user.

Example 1:

[previous impl]:

```
def add(a: int, b: int) -> int:
    """
    Given integers a and b, return the total value of a and b.
    """
    return a - b
```

[unit test results from previous impl]:

Tested passed:

Tests failed:

assert add(1, 2) == 3 # output: -1

assert add(1, 2) == 4 # output: -1

[reflection on previous impl]:

The implementation failed the test cases where the input integers are 1 and 2. The issue arises because the code does not add the two integers together, but instead subtracts the second integer from the first. To fix this issue, we should change the operator from '-' to '+' in the return statement. This will ensure that the function returns the correct output for the given input.

### F.4 TEST CASE GENERATION PROMPT

You are an AI coding assistant that can write unique, diverse, and intuitive unit tests for functions given the signature and docstring.

(input)

## G WEBSHOP PROMPTS

### G.1 ACTING PROMPT

Webshop

Instruction:

i would like a 3 ounce bottle of bright citrus deodorant for sensitive skin, and price lower than 50.00 dollars

[Search]

Action: search[3 ounce bright citrus deodorant sensitive skin]

Observation:

[Back to Search]

Page 1 (Total results: 50)

[Next ] [B078GWRC1J]

Bright Citrus Deodorant by Earth Mama — Natural and Safe for Sensitive Skin, Pregnancy and Breastfeeding, Contains Organic Calendula 3-Ounce

$10.99

[B078GTKVXY]

Ginger Fresh Deodorant by Earth Mama — Natural and Safe for Sensitive Skin, Pregnancy and Breastfeeding, Contains Organic Calendula 3-Ounce

$10.99

[B08KBVJ4XN]

Barrel and Oak - Aluminum-Free Deodorant, Deodorant for Men, Essential Oil-Based Scent, 24-Hour Odor Protection, Cedar & Patchouli Blend, Gentle on Sensitive Skin (Mountain Sage, 2.7 oz, 2-Pack)

$15.95

Action: think[B078GWRC1J and B078GTKVXY are bright citrus deodorant less then 50 dollars. I can check B078GWRC1J first.]

Observation: OK.

Action: click[B078GWRC1J]

Observation:

[Back to Search]

[Prev]

scent [assorted scents][bright citrus][calming lavender][ginger fresh][simply non-scents] size [travel set (4-pack)][3 ounce (pack of 1)][3-ounce (2-pack)] Bright Citrus Deodorant by Earth Mama — Natural and Safe for Sensitive Skin, Pregnancy and Breastfeeding, Contains Organic Calendula 3-Ounce

Price: $10.99

Rating: N.A.

[Description]

[Features]

[Reviews]

[Buy Now]

Action: think[For 3 ounce bottle of bright citrus deodorant for sensitive skin, the item has options 'bright citrus' and '3 ounce (pack of 1)' and seems good to buy.]

Observation: OK.

Action: click[bright citrus]

Observation: You have clicked bright citrus.

Action: click[3 ounce (pack of 1)]

Observation: You have clicked 3 ounce (pack of 1).

Action: click[Buy Now]

## G.2    VALUE FUNCTION PROMPT

Given an item to purchase and a trajectory that aims to buy an item that exactly matches the specification, which corresponds to the ideal score of 1.0, analyze the following trajectory, then at the last line conclude "Thus the correctness score is s", where s is an integer from 1 to 10.

Here are some examples

Webshop

Instruction:

i am looking for dairy free and apple variety pack of chips, and price lower than 30.00 dollars

[Search]

Action: search[dairy free and apple variety pack of chips]

Observation:

[Back to Search]

Page 1 (Total results: 50)

[Next]

[B07HRFJWP8]

Enjoy Life Foods Soft Baked Ovals, Breakfast Bars, Nut Free Bars, Soy Free, Dairy Free, Non GMO, Gluten Free, Vegan, Variety Pack, 4 Boxes (20 Bars Total)

$100.0

[B01KMHY5PG]

Enjoy Life Soft Baked Chewy Bars, Variety Pack, Nut Free Bars, Soy Free, Dairy Free, Gluten Free, 6 Boxes (30 Total Bars)

$21.49

[B008D2X8C4]

Enjoy Life Lentil Chips Variety Pack, Dairy Free Chips, Soy Free, Nut Free, Non GMO, Vegan, Gluten Free, 24 Bags (0.8 oz)

$100.0

Action: think[The first two items are not what I want. The third item is the dairy free and apple variety pack of chips I want.]

Observation: OK.

Action: click[B008D2X8C4]

Observation:

[Back to Search]

[Prev]

flavor name [dill and sour cream][garlic & parmesan][light sea salt][margherita pizza][thai chili lime][variety pack] size [0.8 ounce (pack of 24)][4 ounce (pack of 12)]

Price: $100.0

Rating: N.A.

[Description]

[Features]

[Reviews]

[Buy Now]

Action: think[The item has options 'variety pack' and '0.8 ounce (pack of 24)' and seems good to buy.]

Observation: OK.

Action: click[variety pack]

Observation: You have clicked variety pack.

Action: click[0.8 ounce (pack of 24)]

Observation: You have clicked 0.8 ounce (pack of 24).

Action: click[Buy Now]

Reflection: In this attempt, I was unsuccessful. I accidentally bought a product that was $100, which is more than my budget of $30. Either way, the initial search results were not good. Next time, I will do search["variety pack of chips"] and then check if the results meet the dairy free and the $30 budget constraints. I will continue to refine my searches so that I can find more products.

Thus the correctness score is 5

(input)

## G.3    REFLECTION PROMPT

Given an item to purchase and a trajectory that aims to buy an item that exactly matches the specification, which corresponds to the ideal score of 1.0, analyze the following trajectory, then at the last line conclude "Thus the correctness score is s", where s is an integer from 1 to 10.

Here are some examples

Webshop

Instruction:

i am looking for dairy free and apple variety pack of chips, and price lower than 30.00 dollars

[Search]

Action: search[dairy free and apple variety pack of chips]

Observation:

[Back to Search]

Page 1 (Total results: 50)

[Next]

[B07HRFJWP8]

Enjoy Life Foods Soft Baked Ovals, Breakfast Bars, Nut Free Bars, Soy Free, Dairy Free, Non GMO, Gluten Free, Vegan, Variety Pack, 4 Boxes (20 Bars Total)

$100.0

[B01KMHY5PG]

Enjoy Life Soft Baked Chewy Bars, Variety Pack, Nut Free Bars, Soy Free, Dairy Free, Gluten Free, 6 Boxes (30 Total Bars)

$21.49

[B008D2X8C4]

Enjoy Life Lentil Chips Variety Pack, Dairy Free Chips, Soy Free, Nut Free, Non GMO, Vegan, Gluten Free, 24 Bags (0.8 oz)

$100.0

Action: think[The first two items are not what I want. The third item is the dairy free and apple variety pack of chips I want.]

Observation: OK.

Action: click[B008D2X8C4]

Observation:

[Back to Search]

[Prev]

flavor name [dill and sour cream][garlic & parmesan][light sea salt][margherita pizza][thai chili lime][variety pack] size [0.8 ounce (pack of 24)][4 ounce (pack of 12)]

Price: $100.0

Rating: N.A.

[Description]

[Features]

[Reviews]

[Buy Now]

Action: think[The item has options 'variety pack' and '0.8 ounce (pack of 24)' and seems good to buy.]

Observation: OK.

Action: click[variety pack]

Observation: You have clicked variety pack.

Action: click[0.8 ounce (pack of 24)]

Observation: You have clicked 0.8 ounce (pack of 24).

Action: click[Buy Now]

Reflection: In this attempt, I was unsuccessful. I accidentally bought a product that was $100, which is more than my budget of $30. Either way, the initial search results were not good. Next time, I will do search["variety pack of chips"] and then check if the results meet the dairy free and the $30 budget constraints. I will continue to refine my searches so that I can find more products.

(input)

Reflection:

