# OpenReview forum: "Language Agent Tree Search Unifies Reasoning Acting and Planning in Language Models"
_ICLR.cc/2024/Conference — Submitted to ICLR 2024_

### Official Review · Reviewer_15HT · 2023-10-29

**Soundness:** 3 good
**Presentation:** 3 good
**Contribution:** 3 good
**Rating:** 6
**Confidence:** 4

**Summary:**

The paper proposes LATS, a general framework that unifies the capabilities of LLMs in planning, acting, and reasoning by deliberately constructing trajectories with MCTS and incorporating external feedback. The experiments demonstrate the superiority of LATS by achieving new sota on HumanEval and HotPotQA.

**Strengths:**

- The paper proposes a general framework unifying the capabilities of LLMs in planning, acting, and reasoning

- The paper is well-written and presented clearly.

- The results look promising, achieving new sota on HumanEval and HotPotQA

- The ablation provides some insights into the importance of various strategies when harnessing the power of LLMs.

**Weaknesses:**

- LATS uses a higher computational cost to achieve a better performance, it would be better to add some table or figure explicitly discussing about the tradeoff here.

- It would be clearer to add the exact number of API calls and tokens used etc. for each baseline in the results table since the inference time is not directly comparable to other methods.

- Not much novelty compared to existing LLM prompting techniques.

- It would be interesting to see how LATS performs in real complex planning environments, such as ALFWorld and Minecraft.

**Questions:**

Please address the concerns raised in the Weaknesses.

---

> ### Author Response · Authors · 2023-11-21
> **Response to Reviewer 15HT (Part 1)**
>
> We thank the reviewer for appreciating our work. Below we address each of the reviewer’s concerns.
>
>
> **Q1.LATS uses higher computational cost and requires more results on the tradeoff.**
>
> **1. Theoretically, LATS consumes more token than CoT-SC but has the same token consumption as RAP and ToT.** The token consumption of LATS is higher relative to other simpler baselines (e.g., CoT-SC), as LATS expands with multiple samples every step; the overall token consumption is $n$ times higher than CoT-SC/ReAct with $k$ attempts, where $n$ is the number of samples per expansion. However, as the token cost of other tree-structured search methods (RAP and ToT) are also parameterized by the number of attempts $k$ and the number of nodes expanded $n$, it means that **LATS, RAP, and ToT have the same sample complexity, i.e., same asymptotic token cost**. We summarize the sample complexity in the table below, which is also added to Tab. 7 of our revised submission.
>
> **2. Empirically, LATS expands less nodes upon success than RAP and ToT, which means that it is cheaper.** To count the token consumption more accurately,  we compare the average number of nodes expanded upon success on HotPotQA for LATS, RAP, and ToT. We find that LATS does not only have better performance, but also tends to solve questions with fewer nodes (66.65) sampled than RAP (70.60) or ToT (84.05) with the same sample complexity, indicating a more effective, thus cheaper, search. The result is also shown in the table below.
>
> **3. Even LATS without expansion of multiple nodes (i.e., $n=1$) outperforms other methods with the same sample complexity (CoT-SC, best among multiple runs for ReAct).** To give a fairer comparison between LATS and CoT-SC which does not expand nodes, we also test a version of LATS with $n=1, k=50$, and a version of CoT-SC and ReAct with $k=50*5=250$. We find that ReAct and CoT-SC cannot reach the performance of LATS with lower $k$ and $n=1$, let alone LATS with the same sample complexity ($n=5, k=50$).
>
> Below is the table mentioned above:
>
> Method | Performance on HotPotQA (&uarr;) | Sample Complexity (&darr;) | Average \# of Nodes (&darr;)
> ---|---|---|---
> ReAct (best k = 250) | 0.42 | $O(k)$  |
> CoT-SC (n = 1, k = 250) | 0.40 | $O(k)$ |
> LATS (n = 1, k = 50) | 0.48 | $O(k)$ |
> ToT - ReAct (n = 5, k = 50) | 0.49 | $O(kn)$ | 84.05
> RAP - ReAct (n = 5, k = 50) | 0.54 | $O(kn)$ | 70.60
> LATS (n = 5, k = 50) | 0.61 | $O(kn)$ | 66.65
>
> **Q2. Add the exact number of API calls and tokens used etc. for each baseline in the results table since the inference time is not directly comparable to other methods.**
>
> Thanks for the suggestion. We have added a table showing the general and exact inference cost of all of our tested methods in the updated Tab. 7 in Appendix C of our submission (and also Q1 in the response). We found that 1) with the same sample complexity, our method performs the best, 2) our method with $n=1, k=50$ performs better than other methods with higher $k$, and 3) while LATS has the same sample complexity as ToT and RAP, our method expands less nodes on average upon success, which indicates less token used.
>
> **Q3. Not much novelty compared to existing LLM prompting techniques.**
>
> Although previous work has investigated either search algorithms or self-reflection, the combination of both has not been explored. In contrast, LATS synergistically integrates them, and more importantly, introduces unique and advanced methods to leverage search algorithms and self-reflection, and makes non-trivial adaptation of tree-based search with external feedback. Based on these, we have made significant improvements across multiple testbeds, including 94.4 pass@1 accuracy with GPT-4 on HumanEval and performance comparable to gradient-based fine-tuning methods on WebShop.
>
> **1. Paradigm contributions.** To the best of our knowledge, we are **the first to explore the use of search algorithms with LLM agents for decision-making tasks**. Previous approaches like ReAct and Reflexion cannot sample and select more than one state at every step, limiting exploration in decision-making environments; ToT and RAP do not explore decision-making tasks. It is non-trivial to shift from non-interactive tasks to interactive ones, because an updated node, prompt, and more importantly, search algorithm must be designed to incorporate external input – an achievement we have successfully accomplished. We verify that our proposed method accommodates the external input, even when applied to original non-interactive tasks such as programming.
>
> (Q3 continued in next part)

---

> ### Author Response · Authors · 2023-11-21
> **Response to Reviewer 15HT (Part 2)**
>
> **2. Our LATS improves the use of MCTS.** Compared to RAP, LATS also improves the use of MCTS. In RAP, states are scored with a value function based on self-consistency, while we directly prompt the LM to score the state with justification. Our design is more effective in decision-making, which enables the state to also incorporate an external observation that adds context for a more accurate score. Additionally, LATS samples states through environment interaction, eliminating the need to use the LM as a world model as in RAP. This reduces the risk of error and makes LATS a general LLM framework.
>
> **3. Our LATS improves the use of self-reflection.** Compared to the self-reflection in Reflexion, LATS leverages semantic feedback and failed trajectories in additional ways. These are also added to the context of LM for the value function, thereby refining subsequent state evaluations in addition to the base decision-making.
>
> **4. Our adaptation of search algorithms to external feedback is not trivial, as simply adapting existing tree-based methods, such as ToT and RAP, to external feedback hurts the performance.** In our updated submission, we design a version of ToT and RAP that can incorporate environment feedback in HotPotQA using the ReAct prompt, and find that the new version of ToT and RAP generally performs worse than the reasoning-only setting of HotPotQA (0.55 vs. 0.49 for ToT, and 0.6 vs. 0.54 for RAP). This result is because the information retrieval setting is harder than the multihop question-answering setting, and it highlights that our efforts of adapting search algorithms to decision-making scenarios is non-trivial. The detailed numbers are listed below as well as Tab. 5 of our updated submission:
>
> Method | HotPotQA
> ---|---
> ToT (ReAct) | 0.49
> ToT | 0.55
> RAP (ReAct) | 0.54
> RAP | 0.60
> LATS (DFS) | 0.53
> LATS (No self-reflection) | 0.56
> LATS | 0.61
> LATS (CoT + ReAct) | 0.71
>
>
> **Q4. It would be interesting to see how LATS performs in real complex planning environments, such as ALFWorld and Minecraft.**
>
> We thank the reviewer for the suggestion, though we would like to note that:
>
> **1. ALFWorld is pretty much addressed.** Reflexion has a success rate of above 90% on ALFWorld, which does not leave much room for improvement; on the other hand, Reflexion only achieves a success rate of 35% on WebShop. We chose WebShop because it is a more complex and challenging benchmark while being widely adopted.
>
> **2. While Minecraft is a jewel in the crown for decision-making, it is not a standard benchmark for evaluating prompting methods, and requires too much engineering on the environment itself not relevant to the general decision-making framework.** For example, Plan4MC (Haoqi Yuan et al., 2023) uses a hierarchical RL (PPO+DQN) method to find items in the world, which together only serves as a low-level controller for LLM. Without a widely adopted test suite for LLM alone, the performance of an LLM agent in Minecraft largely relies on the quality of the low-level controller, which is out of our scope of proposing a general framework for LLM decision-making (and other tasks such as reasoning). We do agree with the reviewer that such an environment is an exciting challenge for the next step of LLM decision-making, and added this to the discussion part of the revised submission.
>
>
> We hope that our response has addressed the reviewer’s questions and concerns. We are happy to answer any further questions.
>
> **References:**
>
> Haoqi Yuan et al. Plan4MC: Skill Reinforcement Learning and Planning for Open-World Minecraft Tasks. ArXiv: 2303.16563, 2023.

---

> > ### Comment · Reviewer_15HT · 2023-11-23
> >
> > Thanks for the reply,  I'll keep my score at this stage.

---

### Official Review · Reviewer_7QYU · 2023-10-29

**Soundness:** 2 fair
**Presentation:** 3 good
**Contribution:** 2 fair
**Rating:** 3
**Confidence:** 4

**Summary:**

This paper proposes Language Agent Tree Search (LATS), which hierarchically expands the reasoning path and employs Monte-Calro Tree Search (MCTS) to find the correct reasoning path. Also, to deal with decision making problems, Reflexion mechanism (reflecting past failure episodes and leveraging it in the future rollout) is incorporated. LATS empirically achieves the strong performance in HotpotQA, HumanEval, MBPP, and WebShop, as done in original Reflexion paper.

**Strengths:**

### quality and clarity
- This paper is well-written and easy to follow.

### significance
- The empirical results are strong. 94.4 Pass@1 in HumanEval would be notable results.

**Weaknesses:**

- LATS seems to be the naive combination of existing methods, MTCS from RAP [1] (or ToT [2]) and Reflexion [3], to leverage past (failure) experience. I cannot find a clear difference among those. The originality and significance could be limited from this perspective.
- Evaluation is biased to decision making (HotPotQA \& WebShop). Some reasoning benchmark should be included, such as Game 24, Crossword as done in ToT [2] or GSM8K in RAP [1] to clarify the difference between LATS and ToT/RAP.
- Related to Table 1, I think ToT [2] also incorporates self-refinement process.
- The results of ReAct in Table 5 (WebShop) are lower than the one reported in original paper (Score: 66.6 / SR: 40.0).
- In WebShop, WebGUM [4], a finetuned language model agent, achieves the best performance in SR (Score: 67.5 / SR: 45.0).
- The intention in Figure 4 is ambiguous. Is this a conceptual description of Tree Search?

[1] https://arxiv.org/abs/2305.14992

[2] https://arxiv.org/abs/2305.10601

[3] https://arxiv.org/abs/2303.11366

[4] https://arxiv.org/abs/2305.11854

(Minor Issue)
- In Section 4.2, the definition of $M$ in UCT algorithm is missing.

**Questions:**

- RAP applies at most 20 reasoning iterations for MCTS. This is smaller than LATS (50 iters). Is there any reason for this?
- What is the difference between decision-making and planning in Table 1? I guess both are the same concept.
- What "Memory" means in Table 1?
- How did you measure each metric? Reporting aggregated best among $k=50$ reasoning iterations for MCTS (I guess "best of k" in Table 2)? or reporting the result after $k=50$ reasoning iterations for MCTS? I'm curious about its "learning curve".
- In Table 2 (right), what "CoT+ReAct" means?  In my understanding, ReAct is "CoT" in decision making problem. Also, are there LATS (w/ CoT) in Table 2 (left)?
- On WebShop, it is reported that Reflexion cannot improve the performance as done in ALFWorld. Could you explain what could be the source of improvement of LATS (because LATS employs Reflexion process, too)?

---

> ### Author Response · Authors · 2023-11-21
> **Response to Reviewer 7QYU (Part 1)**
>
> We thank the reviewer for the insightful comments. We are glad that the reviewer found our empirical results significant and the paper well-written. Below we address each of the reviewer’s concerns.
>
> **Q1. LATS is the naive combination of existing methods with limited originality and significance.**
>
> Although previous work has investigated either search algorithms or self-reflection, the combination of both has not been explored. In contrast, LATS synergistically integrates them, and more importantly, introduces unique and advanced methods to leverage search algorithms and self-reflection, and makes non-trivial adaptation of tree-based search with external feedback. Based on these, we have made significant improvements across multiple testbeds, including 94.4 pass@1 accuracy with GPT-4 on HumanEval and performance comparable to gradient-based fine-tuning methods on WebShop.
>
> More concretely, LATS is distinguished from prior work through the following key novelties:
>
> **1. Paradigm contributions.** To the best of our knowledge, we are **the first to explore the use of search algorithms with LLM agents for decision-making tasks**. Previous approaches like ReAct and Reflexion cannot sample and select more than one state at every step, limiting exploration in decision-making environments; ToT and RAP do not explore decision-making tasks. It is non-trivial to shift from non-interactive tasks to interactive ones, because an updated node, prompt, and more importantly, search algorithm must be designed to incorporate external input – an achievement we have successfully accomplished. We verify that our proposed method accommodates the external input, even when applied to original non-interactive tasks such as programming.
>
> **2. Our LATS improves the use of MCTS.** Compared to RAP, LATS also improves the use of MCTS. In RAP, states are scored with a value function based on self-consistency, while we directly prompt the LM to score the state with justification. Our design is more effective in decision-making, which enables the state to also incorporate an external observation that adds context for a more accurate score. Additionally, LATS samples states through environment interaction, eliminating the need to use the LM as a world model as in RAP. This reduces the risk of error and makes LATS a general LLM framework.
>
> **3. Our LATS improves the use of self-reflection.** Compared to the self-reflection in Reflexion, LATS leverages semantic feedback and failed trajectories in additional ways. These are also added to the context of LM for the value function, thereby refining subsequent state evaluations in addition to the base decision-making.
>
> **4. Our adaptation of search algorithms to external feedback is not trivial, as simply adapting existing tree-based methods, such as ToT and RAP, to external feedback hurts the performance.** In our updated submission, we design a version of ToT and RAP that can incorporate environment feedback in HotPotQA using the ReAct prompt, and find that the new version of ToT and RAP generally performs worse than the reasoning-only setting of HotPotQA (0.55 vs. 0.49 for ToT, and 0.6 vs. 0.54 for RAP). This result is because the information retrieval setting is harder than the multihop question-answering setting, and it highlights that our efforts of adapting search algorithms to decision-making scenarios is non-trivial. The detailed numbers are listed in Q2 as well as Tab. 5 of our updated submission.
>
> **Q2. Evaluation is biased to decision-making, and some reasoning benchmark should be included to clarify the difference between LATS and ToT/RAP.**
>
> 1. We would like to clarify that the evaluation is biased because LATS, as its title suggests, is primarily a framework for LM **agents**, and adapting search algorithms to decision-making is one of our core contributions; in a word, we mainly focus on improving performance on decision-making, while maintaining competitive performance on reasoning. The word “unifying” is to describe a single methodology that is able to deal with both reasoning and decision-making, the latter of which (beyond ReAct) is still in its infancy.
>
> (Q2 continued in next part)

---

> ### Author Response · Authors · 2023-11-21
> **Response to Reviewer 7QYU (Part 2)**
>
> Thus, the way we distinguish LATS from ToT/RAP is to evaluate it on benchmarks where an environment feedback is accessible. These planning methods do not natively support external observations, which limits overall performance to the LLM’s existing internal ability. **In fact, trivial handling of external feedback over ToT and RAP even leads to performance drop.** To validate this, we would like to point the reviewer to the updated Tab. 5 in the submission, which we have also added below. Here, we extend ToT and RAP to the decision-making setting by using environmental interactions during sampling with the ReAct prompt, which we denote as **ToT (ReAct) and RAP (ReAct)**. We find that the baselines generally perform worse than the reasoning-only setting of HotPotQA (0.55 vs. 0.49 for ToT and 0.6 vs. 0.54 for RAP), which indicates that the acting-based setting is more challenging and the adaptation of search algorithms to decision-making scenarios is non-trivial.
>
> The performance is listed as follows:
>
> Method | Performance
> ---|---
> ToT (ReAct) | 0.49
> ToT | 0.55
> RAP (ReAct) | 0.54
> RAP | 0.60
> LATS (DFS) | 0.53
> LATS (No self-reflection) | 0.56
> LATS | 0.61
> LATS (CoT + ReAct) | 0.71
>
> 2. We did include results on a reasoning benchmark in Tab.1. This is the standard reasoning setting for HotPotQA as a question-answering benchmark without the external API. We find that while LATS-CoT outperforms ToT, its performance is comparable to RAP. In reasoning-only settings, the advantages of LATS are reduced, as the removal of external feedback also reduces the efficacy of self-reflection and the LM value function. But again, our focus is on maintaining performance on reasoning, while making adaptations for decision-making environments with external feedback.
>
> **Q3. In Table 1, ToT also incorporates a self-refinement process.**
>
> The “self-refine” in Tab. 1 refers to the LM-based generation of semantic feedback. ToT only uses an LM to score states and does not summarize failed trajectories. We have updated the column of “self-reflection” in Tab. 1 to make this distinction clearer.
>
> **Q4. The results of ReAct in Table 5 (WebShop) are lower than the one reported in the original paper.**
>
> The ReAct results in the original paper use PaLM, a proprietary language model that we have no access to. Our experiments use GPT-3.5 because it is available, widely used, and capable of in-context learning. GPT is similar in model scale to PaLM, but has some performance differences.
>
> **Q5. In WebShop, WebGUM achieves the best performance in SR.**
>
> Thanks for pointing this out. We have added WebGUM to Tab. 4 in the updated submission. However, we would like to note that while WebGUM indeed has a higher SR, it uses a different base language model and involves task-specific **fine-tuning**, making it **an unfair comparison** with LATS and other **gradient-free** techniques. Even so, it is noteworthy that on the average score, LATS outperforms WebGUM (75.9 vs 67.5) despite being gradient-free.
>
> **Q6. The intention in Figure 4 is ambiguous.**
>
> Figure 4 is meant to show a qualitative example where LATS can improve over ReAct. To better capture this, we uploaded a new Figure 4 in the updated submission (now Fig. 5). The figure has been moved to Appendix C as Fig. 5 now.
>
> **Q7. In Section 4.2, the definition of M in UCT algorithm is missing.**
>
> Thanks for pointing this out. This should have been N as well. We have fixed the formula in the updated submission.
>
> **Q8. RAP applies less reasoning iterations for MCTS than LATS (20 vs. 50 iters).**
>
> 1. We would like to first clarify that in our original submission, as detailed in the experiment section, we use the same hyperparameters for LATS and RAP – they both use 50 iterations for HotPotQA with each node sampling 5 children; thus, the comparison is fair. Note that, in the RAP’s paper, 20 iterations were used. However, our experiment showed that using 50 iterations improves RAP’s performance, so we adopted this hyperparameter setting than that in RAP’s paper.
>
> 2. Experiment results further suggest that our LATS is more efficient than RAP when it comes to iterations. We test the average number of nodes expanded upon success for LATS, ToT, and RAP on HotPotQA, and find that LATS, while having higher success rate under the same set of hyperparameters, expands less nodes upon success (66.65; RAP uses 70.60 nodes, while ToT uses 84.05 nodes; see Tab. 7 in the updated submission) and thus needs less iteration than RAP and ToT to find the correct answer.
>
> **Q9. What is the difference between decision-making and planning in Table 1? Are they the same concept?**
>
> In LLMs, planning is **defined by algorithm**; it refers to using search algorithms like BFS or MCTS. Decision-making is **defined by problem settings**; it refers to the use of and acting within an external environment. We have updated the caption in Table 1 and changed “decision-making” to “acting” to make this clearer.

---

> ### Author Response · Authors · 2023-11-21
> **Response to Reviewer 7QYU (Part 3)**
>
> **Q10. What does "Memory" mean in Table 1?**
>
> “Memory” refers to the storage of prior states and failed trajectories in an external memory for future use. We have changed the entry to “External Memory” and updated the caption to make this more clear.
>
> **Q11. How did you measure each metric? Reporting aggregated best among k=50 reasoning iterations for MCTS (I guess "best of k" in Table 2)? or reporting the result after k=50 reasoning iterations for MCTS? I'm curious about its "learning curve".**
>
> It is the former; we report the best trajectory out of k. We have also ablated performance over reasoning iterations in Appendix C, Fig. 4 in the updated submission. The results show that LATS has a better learning curve than Reflexion.
>
> **Q12. In Table 2 (right), what "CoT+ReAct" means? In my understanding, ReAct is "CoT" in decision-making problems. Also, are there LATS (w/ CoT) in Table 2 (left)?**
>
> Yes, CoT and ReAct are both base prompt designs for reasoning and decision-making, respectively. The CoT + ReAct version of LATS incorporates both types of base prompts to combine internal reasoning (CoT) and information retrieval (ReAct) strategies for HotPotQA. The API-based setting primarily evaluates the ability of the LM to retrieve information using external tools, while the standard setting evaluates internal reasoning and knowledge, so using both is optimal (Sec 5.1).
>
> Also, we indeed have results of LATS using CoT, which has the same performance as RAP.
>
> **Q13. On WebShop, it is reported that Reflexion cannot improve the performance; then what could be the source of improvement of LATS?**
> In Reflexion, the improvement of Reflexion over ReAct is only through the final summary/feedback generated by the LLM that is added as context. We also find that this summary is less useful in the WebShop difficult environment, but LATS is able to draw additional improvements through the search algorithm. While Reflexion samples one action at every step, LATS samples up to five actions and uses MCTS to optimally explore the state space.
>
> We hope that our response has addressed the reviewer’s questions and concerns. We are happy to answer any further questions.

---

> ### Comment · Reviewer_7QYU · 2023-11-22
>
> Thank you for the detailed response and additional experiments. Here are the remaining concerns I still have:
>
> **Q1.**
>
> I still think the technical novelty is limited because of the combinational nature of the proposed method. For instance, the author mentioned LATS utilizes LLM scoring with justification compared to self-consistency used in RAP. But LLM scoring was adopted in ToT before. Also, the difference between LATS, RAP-ReAct, and ToT-ReAct can be a range of prompt-engineering. I partially agree that it is difficult to unify all. Still, because each component is off-the-shelf and unification in prompting methods can be a prompt-engineering effort, I cannot agree with the significance and technical novelty.
>
> **Q3.**
>
> ToT employs self-refinement in the creative writing experiment, which iteratively incorporates their own feedback for writing.
>
> **Q8.**
>
> What I'd like to point out here is about "its inefficiency", rather than about "fair comparison". 50 trials in decision making tasks while accepting its failures would not be impractical.
>
> **Q12.**
>
> I'm still confused about this. ReAct has "think" action to organize the agent's observations. This could be equivalent to CoT. What is "CoT" in the CoT+ReAct setting?

---

> ### Author Response · Authors · 2023-11-23
> **Response to reviewer 7QYU (1/2)**
>
> We thank the reviewer for the reply. We would like to further clarify the reviewer’s remaining concerns.
>
> **Q1**
> We respectfully disagree with the reviewer about the criteria of novelty. We would like to argue that the simplicity or combinational nature of an approach should not be misinterpreted as a lack of novelty. Instead, simplicity should be regarded as a commendable strength. To support this argument, we refer to the famous perspective from Michael Black on novelty [1]:
>
> > “The simplicity of an idea is often confused with a lack of novelty when exactly the opposite is often true. A common review critique is: the idea is very simple. It just changes one term in the loss and everything else is the same as prior work. However, if nobody thought to change that one term, then it is ipso facto novel. The inventive insight is to realize that a small change could have a big effect and to formulate the new loss.”
>
> > “The idea is obvious because the authors just combined two well known ideas. Obvious is the opposite of novelty. So, if an idea is obvious after you’ve heard it, reviewers quickly assume it isn’t novel. The novelty, however, must be evaluated before the idea existed. The inventive novelty was to have the idea in the first place. If it is easy to explain and obvious in hindsight, this in no way diminishes the creativity (and novelty) of the idea.”
>
> We believe that our work precisely matches this notion of novelty: **It is simple, but is the first to explore a timely problem that leverages search algorithms with LLM agents for decision-making tasks, and demonstrates substantial performance improvement including 86.9 pass@1 accuracy with GPT-3.5 on HumanEval, a 27.6% improvement over Reflexion and higher performance than base GPT-4**. We would respectfully point out that our method makes non-trivial efforts to unify successful past designs and adapt to environments with feedback. In addressing the reviewer’s specific follow-up concerns:
>
> As acknowledged by the reviewer, “I partially agree that it is difficult to unify all.”
> * We have successfully accomplished this difficult task as the first work, and we believe that our LATS framework will lay the foundation for future research in LLM decision-making.
>
> “LATS utilizes LLM scoring with justification compared to self-consistency used in RAP. But LLM scoring was adopted in ToT before”
> * While LLM scoring was adopted in ToT, it is important to note that none of the prior works employed LLM scoring with self-reflection, making our approach novel, effective, and noteworthy within the community.
>
> “The difference between LATS, RAP-ReAct, and ToT-ReAct can be a range of prompt-engineering”
> * We do not feel that using “prompt-engineering” to be a ground for lack of novelty. For instance, ReAct is about the design of prompts based on existing work of chain-of-thought; however, such a simple idea becomes fundamental in the field of LLM decision-making. To reiterate, **our core technical contribution is not based on prompting, but about adapting search algorithms and MCTS to LM agents**.
>
> **References:**
> [1] Novelty in Science. Michael Black. URL https://medium.com/@black_51980/novelty-in-science-8f1fd1a0a143. Accessed Nov. 22nd, 2023.

---

> > ### Author Response · Authors · 2023-11-23
> > **Response to reviewer 7QYU (2/2)**
> >
> > **Q3**
> > We acknowledge this and have updated Table 1. However, we would like to note that unlike the self-reflection in LATS, the self-refinement used in the creative writing experiment of ToT is not general or part of the base ToT design. It is based on the task-specific coherency of the generated passage, which cannot be directly applied to our decision-making environments. In contrast, the self-reflection in LATS describes errors and suggestions and can be applied to any environment.
> >
> > **Q8**
> > We see the reviewer’s concern. The number of trials is adjustable to the problem setting; in programming and WebShop, we in fact used 8 and 30 trials, respectively. We used 50 trials for all methods in HotPotQA to highlight the results of a more effective search process over a higher number of nodes, which we find LATS to also improve efficiency compared to the baselines. This is higher than the 20 trials used for most experiments in RAP because of the larger and more challenging state space in decision-making environments.
> >
> > For additional context, we also provide results for LATS with only 30 trials. We find that  **LATS with 30 trials** is comparable to or outperforming **baseline methods with 50 trials**.
> >
> > Method | Performance | Trials
> > ---|---|---
> > LATS | 0.54 | 30
> > ToT (ReAct) | 0.49 | 50
> > RAP (ReAct) | 0.54 | 50
> > LATS | 0.61 | 50
> >
> >
> >
> > **Q12**
> > The thought portion of ReAct is distinct from the thoughts in CoT. Since CoT does not use observations, the thoughts are independent reasoning steps while in ReAct, the thoughts are based on the observation and what action to take. This means that ReAct and CoT use different strategies on HotPotQA, which is why combining them results in the highest performance. The CoT in CoT+ReAct is the base CoT prompt; we combine the two by switching the base prompt and strategy of LATS to CoT if using ReAct fails.
> >
> > We hope these have addressed the reviewer’s remaining concerns. We are more than happy to discuss, if the reviewer has further questions.

---

> > > ### Comment · Area_Chair_GfPB · 2023-12-05
> > >
> > > Reviewer 7QYU - please take a moment to read the final responses and decide if you would like to keep or change your rating. Thanks.

---

### Official Review · Reviewer_j4Tk · 2023-10-31

**Soundness:** 2 fair
**Presentation:** 1 poor
**Contribution:** 2 fair
**Rating:** 5
**Confidence:** 3

**Summary:**

The paper proposes a MCTS framework which uses LLM as the basic functioning component to perform sequential decision-making tasks.
It achieves strong performance across multiple important benchmarks.

**Strengths:**

The proposed framework is intuitive and easy-to-understand.
It combines modern LLM with classical MCTS algorithm, which is a neat idea.

The empirical results are strong.

**Weaknesses:**

There are some key weaknesses that prevent me from giving an acceptance score.

First, some key technical designs of the proposed framework are not well motivated and seem to be problematic.

E.g., the Abstract says that this method is inspired by model-based RL but the proposed method is model-free. The authors argue that "we can conveniently backup to any state by setting the input..." but, without an environment model, one needs to actually interact with the environment to roll out future steps in order to compute values for possible actions at an earlier step; see Fig-2 and 3. How is that possible in a real application that you could go back up to an earlier step after execution? More importantly, in deployment/inference, using future states of actual interactions does not seem to be a fair comparison with other model-free approaches such as ReAct which doesn't roll out, because it is like an undo move, right?

Second, presentation has major issues. It is easy-to-follow, which is good, but it leaves out much important information so I find it hard to gauge its overall soundness.

E.g., the LATS section is confusing even to a reader familiar with both LLM and MCTS. This section needs a high-level review about MCTS and how it is reshaped with LLM for this framework. Important technical details need to be added: e.g., I didn't find any math formula about backpropagation even after checking within appendices and algorithm boxes.

**Questions:**

Why model-free and why you can still take a previous action after actual roll-out?

---

> ### Author Response · Authors · 2023-11-21
> **Response to Reviewer j4Tk**
>
> We thank the reviewer for the insightful comments. Below we address each of the reviewer’s concerns.
>
> **Q1. The method is inspired by model-based RL, but the proposed method is model-free and can back up; one needs an environment model to back up.**
>
> We thank the reviewer for pointing this out. We would like to clarify the confusion from the following aspects:
>
> 1. Our method is indeed model-free, which we did not explicitly state because it is not common to use this term to describe LM agents. “The inspiration from model-based RL” only lies in the use of MCTS. We thank the reviewer for pointing out this potential source of confusion, and have updated the abstract to make this distinction clearer.
>
> 2. Such statement is true for normal RL environments, but this is not the case for decision-making with LLM because **backing up in LLM decision-making in most settings is simply prompting with previous states of the frozen LLM**.
>
> States in decision-making with LLM come from three sources: task context / prompt (initial state, unchanged during the task), thoughts / reflection (generated by the LLM itself), and external APIs. In LLM benchmarks, the calls of external APIs do not affect each other (e.g., the searches in database are independent in HotpotQA, evaluations are independent in programming, and the xmls can be reset conveniently by navigating to the previous website in Webshop.) Thus, it is easy to **back up in LLM decision-making without an environment model** – we simply store prior history of text and modify the input accordingly, when the search needs to return to an earlier step.
>
> **Q2. Backing up to earlier steps is unrealistic for real applications.**
>
> Though it is true that the ability to back up is a strong assumption in RL environments, **this is not the case for most LLM decision-making settings**, with the reason discussed in Q1. Perhaps in certain large-scale environments like Minecraft this would be harder, so we have acknowledged this in the updated limitations section in Sec. B in the Appendix.
>
> **Q3. The ability to back up is unfair to model-free approaches such as ReAct.**
>
> LATS is indeed model-free, and the ability to back up is fair to methods like ReAct, because LATS uses real environment interactions, which is the same as sampling the next step with ReAct. Such an ability to back up exists in all tasks of interest of all our baselines, including ReAct – however, they overlooked and did not use this capability. Recognizing this fact and leveraging this property is the main motivation of our work.
>
> **Q4. The presentation of LATS is unclear, with review of MCTS and backpropagation procedure missing.**
>
> Thanks for pointing this out. In our updated submission, we have added an overview of standard MCTS, the key adaptations with LM agents, and added the backpropagation formula in Sec. 4.2 as well as the pseudocode in Appendix A.
>
> We hope that our response has addressed the reviewer’s questions and concerns. We are happy to answer any further questions.

---

> > ### Comment · Reviewer_j4Tk · 2023-11-22
> >
> > Thank you for your response. But my key concerns are still not resolved.
> >
> > > backing up in LLM decision-making in most settings is simply prompting with previous states of the frozen LLM.
> >
> > But MTCS requires roll-outs. If you do not have an env model, you have to roll out with real environments. Once you move forward with a real-environment, then how can you back up afterwards? You can not undo what you have done in life, right?
> >
> > > Though it is true that the ability to back up is a strong assumption in RL environments, this is not the case for most LLM decision-making settings
> >
> > I am afraid that it does not sound correct to me.
> >
> > If you use MTCS in an env without using env models, then you need the env to be able to back up---this is a hard constrain, not a strong assumption.
> >
> > And whether it allows a back-up is the property of the env, nothing to do with what kind of decision makers are being used. So why "not the case for most LLM decision-making settings"?
> >
> > > however, they overlooked and did not use this capability.
> >
> > Then you should be clearly upfront about this point in paper.
> >
> > In addition, a fairer comparison will be considering # of steps you need to be correct. But this is not major.
> >
> > With all the issues discussed above, I am not ready to raise the score of this paper. Sorry.

---

> ### Author Response · Authors · 2023-11-23
> **Response to Reviewer j4Tk**
>
> We thank the reviewer for the reply. We would like to further clarify the reviewer’s remaining concerns.
>
> We would like to kindly point out that the **confusion from the reviewer lies in treating the newly emerging environments in which LM agents operates, and which we focus on, as the traditional RL environments – these two types of environments do have distinct properties, especially regarding back-up**. In fact, we would like to further emphasize that explicitly leveraging such back-up capability to propose a principled approach is a key contribution of our work. Below we clarify in more detail.
>
> **Q1. Without environment models, one must roll out with real environments where back-up is not feasible.**
>
> We would like to emphasize that this is not true, at least for the environments we consider. **Our experiments cover API-calling / tool-use, programming, and web-browsing**. These are core benchmarks for LM agents precisely due to their practical value, and LATS can be directly deployed in real-world counterparts. **In WebShop, the environment can be reset to a prior state by changing the xml of the website; and for programming and API-usage, the actions of the LM do not directly affect the environment, so backing up is a matter of changing the prompt**.
>
> **Q2. Being able to back-up is not the property of decision-makers, but the environment.**
>
> This is true, as one could of course use a more “traditional RL agent”, e.g., a LSTM network with a small MLP as the decision head as an actor for the task and back-up. However, **the types of environment that we consider and allow us to back-up is closely related to LM**; for example, HotPotQA and webshop are both recent benchmarks that are considered as NLP tasks; they are largely ignored by the mainstream RL community, and it is only through large language models that the machine learning community gains major progress on solving such tasks. Like we acknowledged in the initial response, the only class of benchmarks where backup is indeed hard are RL environments like AlfWorld and Minecraft, where the agent’s actions permanently change the environment – however, it is important to recognize that this scenario represents only a subset of the environments in which LM agents can be deployed. Thus we feel appropriate to say “LM agents are not constrained by concerns in normal RL”, as the normal RL community does not focus on our task of interest.
>
> **Q3. We should be clearly upfront about prior works overlooking property.**
>
> We would like to emphasize we have already included this.
>
> In our original submission:
>
> 1. In the contribution part ,we write “... construct the best trajectory from sampled actions, enabling more flexible and adaptive problem-solving compared to reflexive prompting methods”, where reflexive means, as we write in the second paragraph, “Despite this strength, these methods are reflexive and fall short of humans' deliberate and thoughtful decision-making characteristics to solve problems. In particular, such methods **fail to consider multiple reasoning paths** or to plan ahead.”
>
> 2.  In the “tree-based search” section of related work, we write “such a problem does not exist for LM tasks as we can conveniently backup to any state by setting the input to be the context and corresponding previous output by the LM.”
>
> In our updated submission:
>
> 1. We made a comparison of our work and prior works in Tab. 1, where we list the use of “planning” as a feature and explain that “We refer to planning as the use of a search algorithm”. We mark this to be true for ToT, RAP and LATS, while false for all other methods.
>
> 2. We add the following text to the introduction of MCTS: “However, such a limitation does not exist for LMs, as we can conveniently reset to any step by simply copy-pasting historical text input. Such a special property is the key motivation of our work.”
>
> We will further emphasize this point in the revision.
>
> **Q4. a fairer comparison will be considering # of steps you need to be correct.**
>
> Below shows the comparison of the sample complexity (asymptotic token consumption, where $n$ is the node expanded every step and $k$ is the number of rollouts), the number of nodes expanded upon success and token consumption, which is also updated in the Tab. 7 of our submission. Compared to ToT and Rap with ReAct prompt, Our method uses less nodes and tokens upon success.
>
> Method | Performance on HotPotQA (&uarr;) | Sample Complexity (&darr;) | Average \# of Nodes (&darr;) | Token Consumption (&darr;)
> ---|---|---|---|---
> ReAct (best k = 250) | 0.42 | $O(k)$  | - | -
> CoT-SC (n = 1, k = 250) | 0.40 | $O(k)$ |  - | -
> LATS (n = 1, k = 50) | 0.48 | $O(k)$ | - | -
> ToT - ReAct (n = 5, k = 50) | 0.49 | $O(kn)$ | 84.05 | 210215
> RAP - ReAct (n = 5, k = 50) | 0.54 | $O(kn)$ | 70.60 | 176500
> LATS (n = 5, k = 50) | 0.61 | $O(kn)$ | 66.65 | 173290
>
> We hope this response has addressed the reviewer's remaining concerns.

---

> > ### Comment · Reviewer_j4Tk · 2023-11-23
> >
> > It is certainly not a confusion about the emerging paradigm of using LLMs as agents.
> > I am super familiar with both classical RL and emerging LLMs.
> >
> > "LLMs as agents" only changes the policy part of RL, but not the environment, is it right?
> >
> > Then why may my arguments be valid in classical RL settings but not in this emerging setting?
> >
> > You said "This is true" to my question about "being able to back up is a property of environment", right?
> >
> > I have been trying to keep my feedback concise and precise.
> >
> > I appreciate that you have written a lot in your response, but the most important thing is actually quite concise here: if back-up is a property of environments, then it doesn't make sense to have a model-free MTCS algorithm for decision-making, because whether or not to do any search is completely determined by the environment, but not your policy / agent. This is the most fundamental problem of this paper, and this is what makes the paper below the acceptance bar.

---

> ### Author Response · Authors · 2023-11-23
> **Response to Reviewer j4Tk**
>
> We thank the reviewer for the quick and concise response. What we are trying to say is that we do not feel "completely determined by the environment" is a fundamental flaw, because our solution already applies to a wide range of LM benchmarks where backup property exists. Some prior methods that focus on only a subset of our tasks of interest have been published at ICLR in the past [1, 2, 3]. We are not developing a universal RL method that applies to all environments, but a language model agent that does decision-making for its corresponding tasks of interest - and we solve them well.
>
> **References:**
>
>  [1] Zhang, S., Chen, Z., Shen, Y., Ding, M., Tenenbaum, J.B., & Gan, C. Planning with Large Language Models for Code Generation. ICLR 2023.
>
> [2] Chen, B., Zhang, F., Nguyen, A., Zan, D., Lin, Z., Lou, J., & Chen, W. CodeT: Code Generation with Generated Tests. ICLR 2023.
>
> [3] Haluptzok, P.M., Bowers, M., & Kalai, A.T. Language Models Can Teach Themselves to Program Better. ICLR 2023.

---

> > ### Comment · Area_Chair_GfPB · 2023-12-05
> >
> > Reviewer j4Tk - please take a moment to read the final responses and decide if you would like to keep or change your rating. Thanks.

---

### Official Review · Reviewer_3Lmb · 2023-10-31

**Soundness:** 3 good
**Presentation:** 2 fair
**Contribution:** 2 fair
**Rating:** 5
**Confidence:** 4

**Summary:**

The paper proposes a new framework called Language Agent Tree Search (LATS) to improve the reasoning and decision-making abilities of large language models (LLMs). Specifically, LATS is a framework that incorporates self-reflection and tree-of-thoughts into LLM-based agent problems. Evaluations across diverse tasks like programming, HotPotQA, and WebShop show LATS effectively harnesses LLM capabilities for reasoning and decision-making.

**Strengths:**

1. The writing is overall clear and easy to follow.
2. The author provides sufficient technical details to understand the LATS framework and reproduce results.
3. The authors evaluate LATS extensively across diverse tasks like programming, HotPotQA, and webshop to demonstrate generality and superiority.

**Weaknesses:**

1. The idea is overall not that novel considering previous work like RAP and Reflextion.

2.  As far as I can see, LATS definitely has much more token consumption compared with other baselines when sampling the same amount of trajectories. I think the author should try to increase the token consumption used in other baselines. For example, report the overall token consumption and try to increase the K set in CoT-SC so it may consume tokens on a similar scale to LATS.

3. The author should try to at least incorporate one baseline with the external environment feedback as the ablation study.

4. The author could provide more ablation studies to analyze the impact of different components, for example, search depth, exploration factor etc..

4. The limitation of LATS is not fully addressed. For example, it seems the current version LATS cannot scale to large-scale problems.

**Questions:**

1. Since the author utilizes LLM itself as the evaluation metric, how do you think of recent works that indicate LLM may not be good at self-critique, for example, Huang et al. 2023 mention that in the experiment of reflection (section 3.1.3 in Huang's paper), they use the correct answer as the criteria to stop the self-correction loop, which is not fair as I think. How do you handle this question during the evaluation of value function in LATS?

2. I understand that LATS leverages the final environment feedback to guide the MCTS search (especially in the backward process). The final result of programming and Webshop is accessible (since you have the simulator), but how can you get the feedback on HotpotQA? Will the HotpotQA environment tell you whether your answer is right or not? If so, it means that you are using a ground-truth answer during the MCTS search process which is completely not reasonable (tbh, this is also related to the phenomenon of using the correct answer as the criteria in my question 1). Could the author elaborate more on what the environment feedback looks like in the HotpotQA environment?

Reference
Huang, Jie, et al. "Large language models cannot self-correct reasoning yet." arXiv preprint arXiv:2310.01798 (2023).

---

> ### Author Response · Authors · 2023-11-21
> **Response to Reviewer 3Lmb (Part 1)**
>
> We thank the reviewer for the insightful comments. Below we address each of the reviewer’s concerns.
>
> **Q1. The idea is not that novel considering previous work like RAP and Reflexion.**
>
> Although previous work has investigated either search algorithms or self-reflection, the combination of both has not been explored. In contrast, LATS synergistically integrates them, and more importantly, introduces unique and advanced methods to leverage search algorithms and self-reflection, and makes non-trivial adaptation of tree-based search with external feedback. Based on these, we have made significant improvements across multiple testbeds, including 94.4 pass@1 accuracy with GPT-4 on HumanEval and performance comparable to gradient-based fine-tuning methods on WebShop.
>
> More concretely, LATS is distinguished from prior work through the following key novelties:
>
> **1. Paradigm contributions.** To the best of our knowledge, we are **the first to explore the use of search algorithms with LLM agents for decision-making tasks**. Previous approaches like ReAct and Reflexion cannot sample and select more than one state at every step, limiting exploration in decision-making environments; ToT and RAP do not explore decision-making tasks. It is non-trivial to shift from non-interactive tasks to interactive ones, because an updated node, prompt, and more importantly, search algorithm must be designed to incorporate external input – an achievement we have successfully accomplished. We verify that our proposed method accommodates the external input, even when applied to original non-interactive tasks such as programming.
>
> **2. Our LATS improves the use of MCTS.** Compared to RAP, LATS also improves the use of MCTS. In RAP, states are scored with a value function based on self-consistency, while we directly prompt the LM to score the state with justification. Our design is more effective in decision-making, which enables the state to also incorporate an external observation that adds context for a more accurate score. Additionally, LATS samples states through environment interaction, eliminating the need to use the LM as a world model as in RAP. This reduces the risk of error and makes LATS a general LLM framework.
>
> **3. Our LATS improves the use of self-reflection.** Compared to the self-reflection in Reflexion, LATS leverages semantic feedback and failed trajectories in additional ways. These are also added to the context of LM for the value function, thereby refining subsequent state evaluations in addition to the base decision-making.
>
> **4. Our adaptation of search algorithms to external feedback is not trivial, as simply adapting existing tree-based methods, such as ToT and RAP, to external feedback hurts the performance.** In our updated submission, we design a version of ToT and RAP that can incorporate environment feedback in HotPotQA using the ReAct prompt, and find that the new version of ToT and RAP generally performs worse than the reasoning-only setting of HotPotQA (0.55 vs. 0.49 for ToT, and 0.6 vs. 0.54 for RAP). This result is because the information retrieval setting is harder than the multihop question-answering setting, and it highlights that our efforts of adapting search algorithms to decision-making scenarios is non-trivial. The detailed numbers are listed under Q3 as well as Tab. 5 of our updated submission.
>
>
> **Q2. LATS has more token consumption than other baselines.**
>
> **1. Theoretically, LATS consumes more token than CoT-SC but has the same token consumption as RAP and ToT.** The token consumption of LATS is higher relative to other simpler baselines (e.g., CoT-SC), as LATS expands with multiple samples every step; the overall token consumption is $n$ times higher than CoT-SC/ReAct with $k$ attempts, where $n$ is the number of samples per expansion. However, as the token cost of other tree-structured search methods (RAP and ToT) are also parameterized by the number of attempts $k$ and the number of nodes expanded $n$, it means that **LATS, RAP, and ToT have the same sample complexity, i.e., same asymptotic token cost**. We summarize the sample complexity in the table below, which is also added to Tab. 7 of our revised submission.
>
> **2. Empirically, LATS expands less nodes upon success than RAP and ToT, which means that it is cheaper.** To count the token consumption more accurately,  we compare the average number of nodes expanded upon success on HotPotQA for LATS, RAP, and ToT. We find that LATS does not only have better performance, but also tends to solve questions with fewer nodes (66.65) sampled than RAP (70.60) or ToT (84.05) with the same sample complexity, indicating a more effective, thus cheaper, search. The result is also shown in the table below.
>
> (Q2 continued in next part)

---

> ### Author Response · Authors · 2023-11-21
> **Response to Reviewer 3Lmb (Part 2)**
>
> **3. Even LATS without expansion of multiple nodes (i.e., $n=1$) outperforms other methods with the same sample complexity (CoT-SC, best among multiple runs for ReAct).** To give a fairer comparison between LATS and CoT-SC which does not expand nodes, we also test a version of LATS with $n=1, k=50$, and a version of CoT-SC and ReAct with $k=50*5=250$. We find that ReAct and CoT-SC cannot reach the performance of LATS with lower $k$ and $n=1$, let alone LATS with the same sample complexity ($n=5, k=50$).
>
> Below is the table mentioned above:
>
> Method | Performance on HotPotQA (&uarr;) | Sample Complexity (&darr;) | Average \# of Nodes (&darr;)
> ---|---|---|---
> ReAct (best k = 250) | 0.42 | $O(k)$  |
> CoT-SC (n = 1, k = 250) | 0.40 | $O(k)$ |
> LATS (n = 1, k = 50) | 0.48 | $O(k)$ |
> ToT - ReAct (n = 5, k = 50) | 0.49 | $O(kn)$ | 84.05
> RAP - ReAct (n = 5, k = 50) | 0.54 | $O(kn)$ | 70.60
> LATS (n = 5, k = 50) | 0.61 | $O(kn)$ | 66.65
>
> **Q3. Try to incorporate baselines with the external environment feedback as the ablation study.**
>
> 1. First, we would like to remind the reviewer that external feedback and observations are already used in ReAct and Reflexion. ReAct is a simple baseline that applies LM to interactive environments, and Reflexion extends ReAct with LM-generated semantic feedback.
>
> 2. Second, we would like to point out that ToT and RAP **do not natively** support feedback. Following the reviewer’s suggestion, we design a version of ToT and RAP that can incorporate environment feedback in HotPotQA using the ReAct prompt, which we denote as **ToT (ReAct) and RAP (ReAct)**. It is important to note that adapting RAP to this setting is non-trivial, as RAP is designed for scenarios where the LM can be repurposed as a world model. To address this issue, we follow the strategy in our proposed LATS and directly use environmental interactions during sampling, but keep the original value function of RAP. The result is shown below (and also Tab. 5 of our updated submission):
>
> Method | HotPotQA
> ---|---
> ToT (ReAct) | 0.49
> ToT | 0.55
> RAP (ReAct) | 0.54
> RAP | 0.60
> LATS (DFS) | 0.53
> LATS (No self-reflection) | 0.56
> LATS | 0.61
> LATS (CoT + ReAct) | 0.71
>
> The results show the following findings:
>
> 1) The DFS version of LATS outperforms ToT (ReAct), indicating the importance of self-reflection in the acting setting;
>
> 2) There is an improvement over RAP (ReAct) with the version of LATS without self-reflection, reflecting the importance of our improved value function in MCTS;
>
> 3) The baselines generally perform worse than the reasoning-only setting of HotPotQA (0.55 vs. 0.49 for ToT and 0.6 vs. 0.54 for RAP), which indicates that the acting-based setting is more challenging and the adaptation of search algorithms to decision-making scenarios is non-trivial.
>
> **Q4. More ablation studies, e.g., search depth, exploration factor, etc.**
>
> Thanks for the suggestion. In the updated submission, we have added Tab. 6 with additional results on search depth and exploration factor for HotPotQA and added the “learning curve” of performance growth with respect to the increase of the number of attempts $k$ on HumanEval in Appendix C (now Fig. 4). The results show that LATS is robust to hyperparameters and scales better with more iterations than Reflexion.
>
> **Q5. Scalability concern of LATS.**
>
> We thank the reviewer for pointing out this possible concern; we have updated our existing limitations section in Sec. B in the Appendix accordingly. However, we would like to point out that 1) Our method is scalable at least on par with RAP/ToT while outperforming them, and 2) **the sample size $n$ for expansion provides a natural, linear trade-off between computational cost and performance**, which can be decreased for large-scale tasks with limited resources. Specifically, when $n=1$, our method is as scalable as ReAct for the same number of trajectories and still performs better as shown in Q2.
>
>
> **Q6. Recent work suggests that LLM may not be good at self-reflection, which may affect the value function in LATS.**
>
> Thanks for bringing the work of Huang et al. 2023 to our attention, though we kindly note that Huang et al. 2023 was released on arXiv after the ICLR submission deadline. We have updated the related work section with this work accordingly. Importantly, the findings presented in Huang et al. 2023 and our work are not contradictory but rather complementary:
>
> 1. Huang et al. 2023 focuses on self-reflection *in the internal reasoning setting*, where there is no external feedback. In contrast, in settings like ours, the LM *has access to an external environment*; for instance, in LATS, the LM has access to test case feedback in programming and API feedback in HotPotQA, so it does not rely solely on itself for self-correction. This is another improvement of LATS over ToT/RAP, as Huang et al. 2023 suggest that LLMs are not good at evaluating their own outputs **without external feedback**.
>
> (Q6 continued in next part)

---

> ### Author Response · Authors · 2023-11-21
> **Response to Reviewer 3Lmb (Part 3)**
>
> 2. Huang et al. 2023 actually supports the argument that self-evaluation of LLM *with external feedback* makes sense. In their discussion section, they provide a list of papers that successfully utilizes self-correction with external feedback, including Gou et al. 2023, Chen et al. 2023, Olausson et al. 2023, Pan et al. 2023, and so on; they also notice that corrections with feedback is a common paradigm for everyday use of LLM. They conclude that “Utilizing this type of feedback, though not perpetually accessible, to assist LLMs in correcting their responses is intuitively beneficial, particularly when the feedback is of high quality.”
>
> **Q7. Getting feedback on the answer correctness of HotpotQA is unreasonable.**
>
> We thank the reviewer for the comment and understand the reviewer’s concern. Below, we would like to clarify the concern from two aspects:
>
> 1. We would like to emphasize that **we match the setting used in Reflexion and ReAct for HotpotQA**. The environmental feedback in HotpotQA consists of all feedback from the API, including whether the answer was correct. It is important to note that we use the same environment for every method, and the use of the correctness labels is the same as that in Reflexion.
>
> 2. The HotpotQA environment is **designed to evaluate information retrieval ability in decision-making** settings. Given the nature of decision-making, it is more reasonable for the environment to provide feedback based on the ground truth or task completion. Note that this might be different from the standard QA setting; to prevent any potential confusion, we have removed the supervised SOTA entry in Tab. 2 of our updated submission to ensure a fair comparison.
>
> We hope that our response has addressed the reviewer’s questions and concerns. We are happy to answer any further questions.
>
> **References:**
>
> Jie Huang et al. Large language models cannot self-correct reasoning yet. arXiv:2310.01798, 2023.
>
> Zhibin Gou et al. Critic: Large language models can self-correct with tool-interactive critiquing. arXiv:2305.11738, 2023.
>
> Xinyun Chen et al. Teaching large language models to self-debug. arXiv:2304.05128, 2023.
>
> Theo X Olausson et al. Demystifying gpt self-repair for code generation. arXiv:2306.09896, 2023.
>
> Liangming Pan et al. Automatically correcting large language models: Surveying the landscape of diverse self-correction strategies. arXiv:2308.03188, 2023.

---

> > ### Comment · Reviewer_3Lmb · 2023-11-22
> >
> > Thanks for the response and revision
> >
> > For Q2.3, what do k,n represent here? In addition, apart from the node visiting number, the actual token consumption is more important (because you mean consume more tokens in each node's expansion.)
> >
> > For Q7, I understand that previous work adopted HotpotQA but it is still not reasonable. Especially considering that in the real world, the QA task won't have any external feedback like the ground-truth label. I would suggest the author try to replace this experiment with some other ones.

---

> > > ### Author Response · Authors · 2023-11-23
> > > **Response to reviewer 3Lmb**
> > >
> > > We thank the reviewer for the reply. We would like to further clarify the reviewer’s remaining concerns.
> > >
> > > **Q2.3**
> > >
> > > For our experiments, $k$ represents the maximum number of trajectories sampled during search, each of which includes $n$ nodes sampled at every depth level.
> > > We agree that actual token consumption would be a more accurate assessment, but the prompt length is generally comparable across methods. LATS has a longer prompt after self-reflection by around a hundred tokens; we have updated the table with the exact amount of tokens sampled to answer a question successfully on average. Due to the reduced number of total nodes sampled, LATS is generally still cheaper than other search methods.
> > >
> > > Method | Performance on HotPotQA (&uarr;) | Sample Complexity (&darr;) | Average \# of Nodes (&darr;) | Token Consumption (&darr;)
> > > ---|---|---|---|---
> > > ReAct (best k = 250) | 0.42 | $O(k)$  | - | -
> > > CoT-SC (n = 1, k = 250) | 0.40 | $O(k)$ | - | -
> > > LATS (n = 1, k = 50) | 0.48 | $O(k)$ | - | -
> > > ToT - ReAct (n = 5, k = 50) | 0.49 | $O(kn)$ | 84.05 | 210215
> > > RAP - ReAct (n = 5, k = 50) | 0.54 | $O(kn)$ | 70.60 | 176500
> > > LATS (n = 5, k = 50) | 0.61 | $O(kn)$ | 66.65 | 173290
> > >
> > > **Q7**
> > >
> > > We understand the reviewer’s concern, but we would like to emphasize that our HotPotQA setting is not meant to evaluate the use of an LLM as a standard QA system, but if it can make use of an environment to complete a decision-making task. And we choose the current setting with multiple precedents to align with the prior works, and show that LATS indeed outperforms ReAct-based versions of ToT and RAP.
> > >
> > > Meanwhile, as pointed out by the reviewer, previous works adopted HotPotQA and we adhered to this practice for ease and fair comparisons. We agree with the reviewer that designing improved benchmarks is a general problem in the community, which is an excellent direction for future research. We have also updated our submission to reflect this accordingly.
> > >
> > > We hope this has addressed the reviewer’s remaining concerns.

---

> > > > ### Comment · Area_Chair_GfPB · 2023-12-05
> > > >
> > > > Reviewer 3Lmb - please take a moment to read the final responses and decide if you would like to keep or change your rating. Thanks.

---

### Author Response · Authors · 2023-11-21
**General Response to Reviewers**

We thank all the reviewers for their constructive feedback on our work. We are delighted to see that the reviewers have carefully evaluated our work, given many valuable feedbacks, and highlight that

Our proposed solution is neat, general, intuitive and easy to understand (by reviewers j4Tk and 3Lmb) with clear presentation (by all reviewers), and strong (by all reviewers) and extensive (reviewer 3Lmb) empirical results and ablations.

We address all the reviewers’ concerns in the individual response. In addition, we have updated our submission pdf and marked modified parts red for better illustration of changes. In particular, we have provided experimental results on the following:

1. Computational cost analysis between LATS and baseline methods,
2. Ablations on search depth, exploration factor, and performance over iterations, and
3. Comparisons with baseline methods with standardized cost.

We have also updated the limitation according to the reviewers’ suggestion, add a subsection in preliminary about MCTS and why it is applied in our solution, and update pseudocode to include more implementation details (requested by reviewer j4Tk).

---

### Public Comment · ~Dong_HUANG4 · 2023-12-06
**Paper's evaluation metric for the code generation**

**We use k = 8 iterations, set the number of generated tests at 4,
and sample n = 5 solutions during expansion. After the search is completed, we select the solution
with the highest value and evaluate it on the real test suite for the pass@1 accuracy evaluation.**

Based on the statement, I think the metric used should be pass@5 not pass@1?

---

### Meta-Review · Area_Chair_GfPB · 2023-12-05

**Metareview:**

The paper introduces a new framework that significantly improves the reasoning and decision-making abilities of large language models (LLMs). The proposed framework combines self-reflection, tree-of-thoughts, and Monte-Carlo Tree Search (MCTS) to leverage the strengths of LLMs in planning, acting, and reasoning. This results in a more deliberate and adaptive problem-solving mechanism that outperforms existing methods on diverse tasks including programming, HotPotQA, and WebShop. In general the reviewers appreciate the strong empirical results on targeted tasks and the effectiveness in harnessing LLM capabilities for reasoning and decision-making. The reviewers are concerned about (1) high compute cost, (2) novelty (this work can be seen as a combination of existing LLM prompting techniques), (3) applicability to environments that don't support backup, and (4) applicability to environments that don't have ground-truth feedback. While not putting much weight on (1) and (2), the AC thinks the limitation and additional access to privilege information regarding (3) and (4) should be emphasized much more in the paper, for fair comparison to prior work (e.g. in Table 1-5, introduction, conclusion). Thus, the AC recommends rejecting this work. The authors are encouraged to incorporate these feedback and resubmit to a future venue.

**Justification For Why Not Higher Score:**

Lack of clear description of the limitation and problem setting assumption in the paper, which could lead to a false impression that this is a strictly better method than prior art in all circumstances.

**Justification For Why Not Lower Score:**

N/A

---

### Decision · Program_Chairs · 2024-01-16

Reject